# DOES LEARNING THE RIGHT LATENT VARIABLES NECESSARILY IMPROVE IN-CONTEXT LEARNING?

## ABSTRACT

Large autoregressive models like Transformers can solve tasks through in-context learning (ICL) without learning new weights, suggesting avenues for efficiently solving new tasks. For many tasks, e.g., linear regression, the data factorizes: examples are independent given a task latent that generates the data, e.g., linear coefficients. While an optimal predictor leverages this factorization by inferring task latents, it is unclear if Transformers implicitly do so or if they instead exploit heuristics and statistical shortcuts enabled by attention layers. Both scenarios have inspired active ongoing work. In this paper, we systematically investigate the effect of explicitly inferring task latents. We minimally modify the Transformer architecture with a bottleneck designed to prevent shortcuts in favor of more structured solutions, and then compare performance against standard Transformers across various ICL tasks. Contrary to intuition and some recent works, we find little discernible difference between the two; biasing towards task-relevant latent variables does not lead to better out-of-distribution performance, in general. Curiously, we find that while the bottleneck effectively learns to extract latent task variables from context, downstream processing struggles to utilize them for robust prediction. Our study highlights the intrinsic limitations of Transformers in achieving structured ICL solutions that generalize, and shows that while inferring the right latents aids interpretability, it is not sufficient to alleviate this problem.

## 1 INTRODUCTION

Recent achievements of large language models (LLMs) showcase the Transformer architecture's (Vaswani et al., 2017) capacity to extend beyond predictive modeling and solve novel tasks at inference time (Bubeck et al., 2023). However, several papers (Tang et al., 2023; McCoy et al., 2019; 2023) find that rather than modeling the tasks themselves, Transformers rely on shortcuts that risk generalizing poorly to new datasets and tasks. In this work, we aim to shed light on whether learning the true task parameters over shortcuts is sufficient to aid generalization. In particular, we consider the specific case of in-context learning (ICL): the ability of Transformers to leverage demonstrations within their input sequence to adapt to novel queries, effectively learning from contextual cues. ICL underpins much of the capabilities of modern LLMs, including prompt engineering and chain-of-thought, but is difficult to study in LLMs directly due to the multitude of factors that affect performance and lack of control over the training data. To disentangle the role that task latents play in ICL from other aspects that influence large, pre-trained autoregressive models, we therefore introduce controlled experimental settings that involve training Transformers from scratch on tasks that are sufficiently complex, but where the relevant latent variables are well-understood.

Many tasks naturally admit a parametric approach to ICL that breaks the prediction mechanism into two parts: 1) inferring the task-dependent latent variables from the context, and then 2) using them to make predictions on novel queries. For instance, in a linear regression task, a model could first try to infer the underlying weight vector used to generate the data, and subsequently use these inferred weights to make predictions for query points. Another viable methodology, closer in spirit to non-parametric approaches, is to directly compare the query point with the context through a kernel-based mechanism – weighting predictions based on some learned measure of distance between the different points. The two paradigms, parametric vs. non-parametric, reflect tradeoffs between the potential for better generalization or increased flexibility (Russell & Norvig, 2010).

Figure 1: We compare the benefits of the implicit (*left*) and the explicit (*right*) model. Explicit models disentangle context aggregation and prediction into two separate functions with an inductive bias for inferring generative latent variables in order to solve the task. Implicit models are more expressive, but can learn non-parametric shortcut solutions that bypass latent variable inference.

Recent studies show that Transformers are indeed able to solve ICL tasks through the parametric mechanisms in some cases (Hendel et al., 2023; Todd et al., 2024). However, in most cases, evidence shows that they instead rely on non-parametric mechanisms where the prediction is made by directly comparing the query point to exemplars in the context (Wang et al., 2023; Han et al., 2023), a mechanism very much related to induction heads (Olsson et al., 2022). This could be explained by the fact that the functional form of attention operations is almost identical to that of kernel regression (Tsai et al., 2019), making such solutions more natural for Transformers to express (Zhou et al., 2023). These solutions are considered statistical shortcuts since they might not be able to generalize to OOD contexts and queries – for e.g., learning the actual linear predictor for linear regression can generalize to any distribution over training and test points, but nearest-neighbour based interpolation might not.

In this paper, we aim to test the hypothesis that a limiting factor of ICL in Transformers is their tendency to prefer non-parametric shortcuts over more structured inference. To do so, we perform a thorough analysis through tasks for which latent mechanisms are well understood, and systematically analyse the impact of latent task representation on generalization. We minimally modify the Transformer architecture to prevent such non-parametric shortcuts and compare the OOD performance of the resulting model to that of a traditional Transformer on a large array of ICL tasks. We call this altered architecture an *explicit model* by virtue of its inductive bias of explicitly extracting structured latent variables to solve the tasks, and we call the traditional Transformer architecture an *implicit model*. Specifically, the explicit model prevents the query from directly attending to demonstrations in the context by introducing a bottleneck between the processing of the context and the query (see Figure 1), similar to a conditional neural process (Garnelo et al., 2018a).

We find that the explicit model does not outperform the implicit one on OOD data, challenging the aforementioned hypothesis that avoiding non-parametric solutions would enhance generalization. Our investigation into this lack of improvement reveals that the issue often lies in the explicit model's prediction function, which is tasked with leveraging the inferred latent variables for downstream predictions on the query. Our controlled experiments and analysis on the interpretable nature of the bottleneck revealed strong evidence that while the explicit model often extracts relevant task latents, these are not properly utilized by the prediction function.

While on one hand, our research demonstrates that using a simple bottleneck in a Transformer can improve interpretability and explicitly extract task-relevant latent variables, it also suggests that the limitations of Transformers in learning more structured and generalizable ICL solutions are not solely due to non-parametric shortcuts that skirt latent variable inference, but due to more fundamental architectural limitations.

In sum, our contributions are:

- Formalizing an experimental framework to evaluate the hypothesis that parametric ICL solutions generalize better out-of-distribution.
- Analyzing the benefits, or lack thereof, of inferring the true latents explicitly.
- Identifying shortcomings in the prediction function and the downstream utilization of learned latents in Transformers, which leads to poor generalization.

## 2 NOTATION

Throughout the paper, we denote datasets with the symbol $\mathcal{D}$ which consists of a set of observations with inputs denoted via $\boldsymbol{x} \in \mathcal{X}$ and their corresponding outputs as $y \in \mathcal{Y}$. A task is defined by a functional mapping $g : \mathcal{X}, \mathcal{Z} \rightarrow \mathcal{Y}$ which maps observations $\boldsymbol{x}$ to labels $y$ through some latents or parameters $\boldsymbol{z}$, eg. $y = \boldsymbol{z}^T \boldsymbol{x}$ for a linear regression task, or $y \sim \mathcal{N}(\cdot; \boldsymbol{z}^T \boldsymbol{x}, \sigma^2)$ for its stochastic counterpart. To ease readability, we will reserve $\boldsymbol{x}_* \in \mathcal{X}$ for the query point, i.e. the test time observation we want to generalize to, and $y_* \in \mathcal{Y}$ its corresponding target. Finally, $\psi$ denotes the parameters of context aggregation component of explicit model, which inputs the dataset $\mathcal{D}$ and infers the corresponding parameters $\boldsymbol{z}_\psi(\mathcal{D})$, and $\gamma$ the parameters of the prediction model which given a query $\boldsymbol{x}_*$ and parameters $\boldsymbol{z} \in \mathcal{Z}$, provides the prediction. For the implicit model, these operations are subsumed into a single model, with parameters $\varphi$.

## 3 IMPLICIT VS. EXPLICIT INFERENCE

We look at ICL in the context of algorithmic problems where the task is to predict the target $y_*$ from a query point $\boldsymbol{x}_*$ when provided with some context examples $\mathcal{D} = \{(\boldsymbol{x}_i, y_i)\}_{i=1}^n$, sharing a common underlying structure defined by the task latent $\boldsymbol{z}$ and a functional form $g$. The goal of ICL is to learn a function that can utilize the context set $\mathcal{D}$ to provide predictions for new query points $\boldsymbol{x}_*$. To achieve this, the model is trained on different draws of context sets $(\mathcal{D}_1, \mathcal{D}_2, ...)$ which share the same underlying functional mapping $g : \boldsymbol{x}, \boldsymbol{z} \rightarrow y$ but different realizations of the latent $\boldsymbol{z}$, for example $g(\boldsymbol{x}, \boldsymbol{z}) = \boldsymbol{z}^T \boldsymbol{x}$ could be a linear regression system shared across different contexts $\mathcal{D}_1, \mathcal{D}_2, ...$, but the underlying latents could be different, i.e. $\mathcal{D}_1$ is generated from $\boldsymbol{z}_1$ while $\mathcal{D}_2$ from $\boldsymbol{z}_2$, similar to Von Oswald et al. (2023). We emphasize that in this setup, we are not training models to do next-token prediction as is done in language modeling; instead, given a fixed context $\mathcal{D}$ that includes $n$ samples, we are attempting to make a prediction on a single novel query $\boldsymbol{x}_*$. We therefore do not use a causal Transformer, and we allow all tokens to attend to each other.

Often, ICL solutions are learned via maximum likelihood, i.e.

$$\arg\max_\varphi \quad \mathbb{E}_{\mathcal{D}, \boldsymbol{x}_*, y_*} \left[\log p_\varphi(y_* | \boldsymbol{x}_*, \mathcal{D})\right] \tag{1}$$

where $p_\varphi$ represents the Transformer model and $\mathcal{D}$ is sampled from the parametric family defined through $g$. Thus, the transformer model $p_\varphi$ must not only learn the form of the prediction function $g$, but also how to efficiently aggregate information from the context $\mathcal{D}$ to infer $\boldsymbol{z}$ for downstream predictions on $\boldsymbol{x}_*$. Thus, this general framework can be naturally decomposed into two distinct parts.

**Context Aggregation.** This component deals with inferring the task-dependent latent variables from the in-context examples such that the downstream prediction becomes conditionally independent of the context, i.e. inferring $\boldsymbol{z}$ from $\mathcal{D}$ such that $p(y_* \mid \boldsymbol{x}_*, \boldsymbol{z}, \mathcal{D}) = p(y_* \mid \boldsymbol{x}_*, \boldsymbol{z})$.

**Predictive Modeling.** This component refers to the process of estimating the predictive function that leverages context $\mathcal{D}$ to infer $y_*$ from a query $x_*$. In the above example, it refers to learning the functional mapping $g$ once $\boldsymbol{z}$ has been extracted from context aggregation.

As discussed, Transformers do not have a clear incentive to make this explicit separation of context aggregation and predictive modeling. Instead, given context $\mathcal{D}$, they implicitly and jointly model both the function $g$ along with $\mathcal{D}$-dependent latent variable $\boldsymbol{z}$ inference to directly provide predictions for the query point $\boldsymbol{x}_*$, in contrast to separately estimating $g$ and an explicitly factorized $\boldsymbol{z}$. Thus, in order to enforce explicit representation of $\boldsymbol{z}$, we propose the simplest architectural modification where the query $\boldsymbol{x}_*$ cannot directly attend to the context, and the latent task representation is forced to summarize the context efficiently. Formally, we compare the following two models, which are illustrated in Figure 1.

**Implicit Model.** This refers to the traditional in-context learning computation performed by Transformer models. In this setup, given the set of observations $\mathcal{D}$ (context) and a query point $\boldsymbol{x}_*$, the prediction $y_*$ is modeled directly as $p_\varphi(y_* | \boldsymbol{x}_*, \mathcal{D})$, where $p_\varphi$ is defined using a standard Transformer with parameters $\varphi$ and is tasked with modeling both context aggregation and predictive modeling.

**Explicit Model.** This represents the architectural variation which minimally modifies the Transformer architecture by separating context aggregation and predictive modeling. It first constructs a task representation $\boldsymbol{z}_\psi(\mathcal{D})$ using the set of observations $\mathcal{D}$ and a context model $\boldsymbol{z}_\psi$ with parameters $\psi$ (*context aggregation*) and another network $p_\gamma$ to make a prediction for a new point $\boldsymbol{x}_*$ (*predictive*

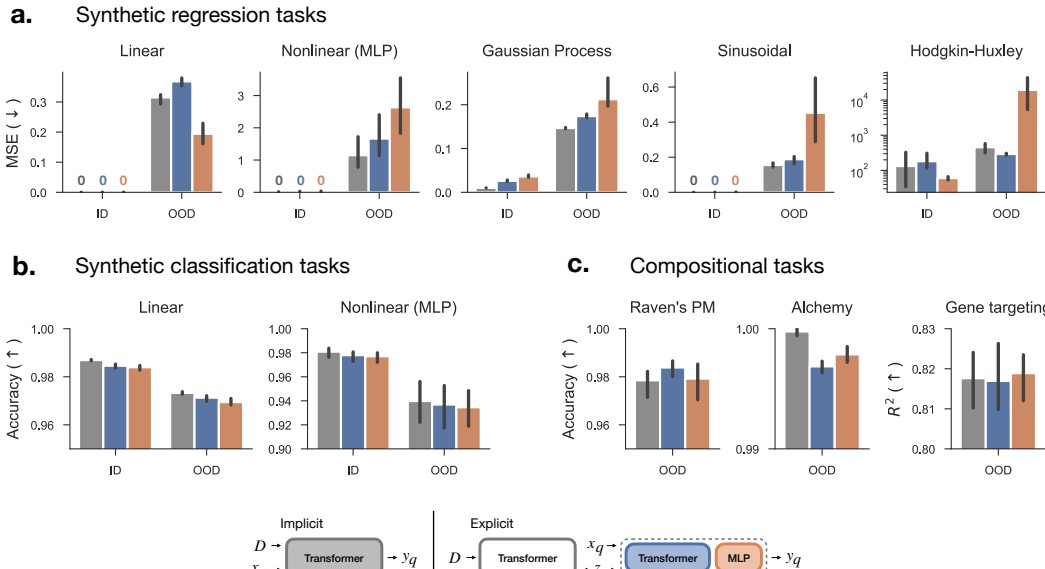

Figure 2: Comparison of implicit and explicit models in-distribution (ID) and out-of-distribution (OOD) across various domains: (a) synthetic regression, (b) classification, and (c) compositional generalization tasks. Implicit models are in shown gray, explicit models with Transformer prediction in blue, and with MLP prediction in orange. Further details about tasks is provided in Appendix B.

*modeling*) as $p_\gamma(y_*|\boldsymbol{x}_*, \boldsymbol{z}_\psi(\mathcal{D}))$. A key insight is that the *task latents* are invariant to the queries when modeling prediction. The context model is implemented with a Transformer $\boldsymbol{z}_\psi$ with weights $\psi$, and for the prediction function $p_\gamma$, we study both Transformers and MLPs with weights $\gamma$. Importantly, the output of the context model $\boldsymbol{z}_\psi(\mathcal{D})$ is a fixed-size vector with much lower dimensionality than the full context $\mathcal{D}$. This information bottleneck prevents the query $\boldsymbol{x}_*$ from attending directly to the context as in standard Transformers; instead, the context model must summarize $\mathcal{D}$ into underlying generative factors, thus ruling out potential shortcut solutions that bypass latent variable inference.

**Implicit vs. Explicit.** Assuming Transformers do in fact favour shortcut-based solutions, we first hypothesize when each setup should perform better given different task characteristics. If the data is generated with a linear model (i.e. $y = \boldsymbol{z}^T\boldsymbol{x}$), the right predictor can be precisely described using the weight vector $\boldsymbol{z}$, making the explicit model better suited. In contrast, if the data is generated with a Gaussian Process (GP), the implicit model should be superior since by construction query prediction computes similarities with all points in the context. In this case, the task latents of GP-based data with RBF kernel is infinite dimensional (i.e. a point in function space), which cannot be captured in the finite-dimensional bottleneck of the explicit model. In general, we should expect the explicit model to be superior when the underlying true model is parametric and low-dimensional, but in case of a non-parametric or very high dimensional parametric model, the implicit model should be better.

Finally, we note that our aim is *not* to construct the best possible explicit model architecture – indeed, more sophisticated ones already exist based on amortized Bayesian inference (Garnelo et al., 2018b; Mittal et al., 2023). Instead, we are interested in investigating potential inductive biases for ICL by *minimally* modifying the standard Transformer architecture to remove certain shortcuts from the space of possible solutions. We leave the design of more performant architectures for future work.

We further refer the readers to Appendix A for a detailed discussion of related work.

## 4 EXPERIMENTS

Our goal is to use simple tasks that capture the key elements of ICL applications to tease apart the effects of implicit and explicit models on generalization, both in-distribution (ID) and out-of-distribution (OOD).

**Task Setup.** We conduct experiments across a variety of settings that admit task latents, from synthetic regression and classification to reasoning-based problems. For reasoning tasks that require

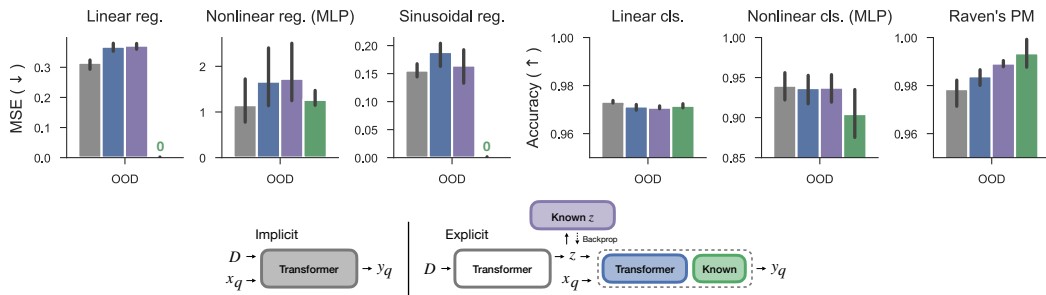

Figure 3: Performance comparisons on a subset of tasks where the true latent variable $z$ and prediction function $g$ are known. Implicit models are in shown gray and explicit models with Transformer prediction are in blue. Models in purple are trained with an auxiliary loss to predict the true latent variable. Models in green are defined using the true prediction function. Using the known prediction function leads to significantly better OOD performance.

compositional knowledge, we consider Raven's Progressive Matrices (Raven's PM) (John & Raven, 2003), Alchemy (Wang et al., 2021), and Gene Targeting (Norman et al., 2019). A brief description of our tasks is provided below, with a more thorough explanation in Appendix B.

*Regression Tasks.* We consider different data-generating processes, e.g., linear: prediction $z^T x$ and latents $z$, nonlinear (MLP): prediction with a neural network $g(x, z)$ and latents as its weights, sinusoidal: prediction as a sum of sinusoids with different frequencies with latensts as their amplitudes, etc.

*Classification Tasks.* Akin to the regression problems, we consider a linear and nonlinear (MLP) prediction for classification using an additional sigmoid activation on the output.

*Raven's Progressive Matrices.* A pattern-completion task used in IQ tests where individual object attributes evolve according to different rules. The task is to complete a sequence of frames such that the underlying rule, which is the latent variable and can be based on colors, shapes, etc., is maintained.

*Alchemy.* Here, an unknown latent causal graph describes how different stones and potions interact to give rise to new stones. The task is to infer the next stone given a history of transitions.

*Gene Targeting.* It represents a real-world dataset of targeted gene knockouts and subsequent observations of gene expressions across many cells. The underlying latent variable is the set of genes that were knocked out in a given experiment, and the task is to infer the gene expressions of certain cells in an experiment given those of other observed ones.

*Reusable Modular Mixture of Experts (MoE).* In this task, we apply a sequence of operations $g_l$ on the input $x$, where the choice of expert applied at layer $l$ is governed by a categorical latent $z_l$. In particular, we apply five operations sequentially leaduing $y = g_{z_5} \circ g_{z_4} \circ \ldots \circ g_{z_1}(x)$. This represents a reusable mixture of experts system with a hierarchical and compositional decomposition.

**Training and Evaluation.** Tasks based on regression utilize the mean-squared error loss, while those based on classification use the cross entropy loss for training. Model architecture details are provided in Appendix C.1. For ID evaluation, we consider the underlying task latent $z$, context samples $\mathcal{D}$, and queries $x_*$ to be sampled from the same distribution as during training. The challenge in this case is simply to generalize from finite data. For OOD evaluation, we test two different cases depending on the kind of task. For our synthetic regression and classification tasks, the task latent $z$ and context samples $\mathcal{D}$ are sampled from the same distribution as at training time, but the queries $x_*$ are sampled from a Gaussian distribution with higher ($3\times$) standard deviation, thus testing a form of out-of-domain generalization. For our reasoning-based problems, we evaluate on task latents $z$ that weren't seen at training. The task latent in each of these reasoning-based problems is composed of parts (e.g., in the Gene Targeting experiment, the latent is a binary vector of targeted genes), which allows us to test a form of *compositional* generalization (Wiedemer et al., 2023) in which all parts have full marginal support at training time, but novel combinations of those parts are evaluated at test-time.

For all tasks, implicit and explicit Transformer models were trained from scratch over a distribution of tasks latents, and we always control for the number of network parameters to provide a systematic comparison between the implicit and explicit models. To better understand the shortcomings of

different models, we also compare with privileged oracles (known decoder – using ground-truth $g$ function, and known latent variable – using an auxiliary loss that includes the ground-truth $z$).

**Explicit models do not outperform the implicit models.** The first evaluation setting that we considered was the ID performance. In this case, we should expect both implicit and explicit models to perform equally well, even if implicit models learn shortcuts rather than ground-truth task latents. This is because those shortcuts are tuned to minimize prediction error within the same data distribution that is being evaluated. Across all our tasks, the results indeed confirmed this prediction. Specifically, during ID evaluation, all models were capable of making highly accurate predictions (Figure 2). While the performance of the implicit model was generally slightly better than that of the explicit models, the benefits were marginal. Effectively, all models solved the tasks similarly well.

While we expected that ID evaluation would be insufficient to demonstrate potential benefits of the explicit models, we expected to see differences in OOD settings. Both implicit and explicit models are sufficiently expressive to fit the training distribution. However, if an explicit model successfully learns the true task latents that generated the data while an implicit model learns statistical shortcuts that are specialized to the training distribution, we should expect the explicit model to generalize better

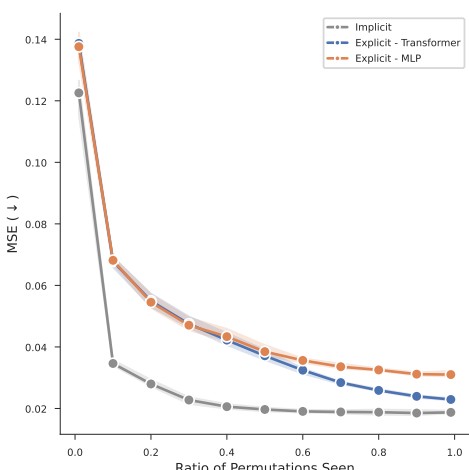

Figure 4: We conduct experiments on reusable modular MoE task where we only train on a subset of combinations of experts, shown on the X-axis. Our results indicate that across different percentages of combinations seen during training, the implicit model consistently outperforms the explicit ones in such a compositional generalization task.

OOD. As a reminder, for the synthetic regression and classification tasks in Figure 2 (a, b), OOD evaluation was done by sampling $x_*$ from a normal distribution with wider standard deviation than was used at training ($3\times$), effectively evaluating if the models could extrapolate to points further out along their domain despite only being trained within a narrow distribution near the origin. For the compositional tasks in Figure 2 (c), we instead evaluated OOD performance by only training on certain configurations of the true latent variable $z$ while evaluating on unseen ones. Importantly, at training time the models were shown data that included every possible value for each component of $z$, but not every possible combination of these values were seen, thus evaluating a form of compositional generalization (Wiedemer et al., 2023). We additionally also conduct a similar experiment on the reusable modular MoE task in Figure 4 which highlights that across different data starvation regimes where only a subset of combinations are seen during training, implicit models still consistently outperform explicit ones even when evaluation is on all combinations.

Surprisingly, and counter to our predictions above, we found that the explicit model provided no significant benefit in OOD settings. In synthetic regression tasks shown in Figure 2 (a), all models failed to generalize and obtained substantially worse performance than during ID evaluation, with the implicit model actually being the one that had a slightly lower degradation in performance. In classification and compositional tasks shown in Figure 2 (b, c), all models generalized fairly well OOD and with similar performance. In summary, explicit models appear to provide no benefit across our tasks, both in terms of ID and OOD performance.

If the explicit model did learn the right latent variables in the bottleneck, it essentially implies that either the implicit model learns benign shortcuts (if at all) or that learning the right latent variables is not sufficient to improve generalization, both ID or OOD. In the following results, we see that the explicit model does indeed learn the right task latents.

**Explicit models learn to infer the correct latent variable, but not how to use it.** Why didn't the explicit model provide any benefit? Our initial hypothesis was that the implicit model could be susceptible to learning shortcuts that are sufficient to reduce the training loss and easy to express using self-attention between the query $x_*$ and context $\mathcal{D}$. By summarizing the context in a bottleneck $z_\psi$, the explicit model should instead be forced to extract the true latent variable in order to minimize

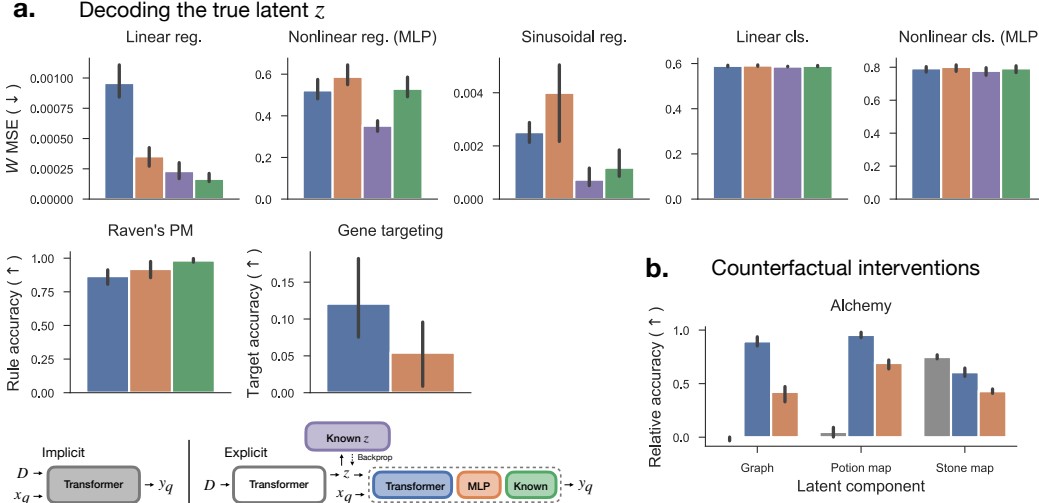

Figure 5: Explicit models are interpretable as the bottleneck allows us to (a) linearly decode the true latent, and (b) intervene on it to obtain correct counterfactual predictions. Implicit models are shown in gray, explicit models with Transformer prediction in blue, and with MLP prediction in orange. Models in green use the true prediction function $g$, while models in purple use an additional auxiliary loss based on ground-truth latents. To evaluate decoding performance in (a), we also build a baseline by linearly decoding the true latent directly from concatenated context examples, which yielded significantly worse performance than decoding from the bottleneck. Baseline performances in units of the plots are – Linear regression: $0.49$, Nonlinear regression (MLP): $0.94$, Sinusoid regression: $0.33$, Linear classification: $0.86$, Nonlinear classification (MLP): $0.97$, Raven's PM: $0.5$, and Gene targeting: $0.0$. In (b), the "Relative accuracy" takes in account the baseline (details in C.4).

the training loss, thus learning a solution that is closer to the actual data-generating process. There are then two possible explanations for our results: (1) the explicit models did not learn to extract the true latent variable despite inductive biases to do so induced by the bottleneck, or (2) they did extract the true latent variable but did not learn to use it in the correct way. We performed several experiments to test these two possibilities, and found strong evidence for the second.

To test whether or not the explicit models failed because they did not extract the correct latent variable, we first attempted to encourage them to do so more directly by training an additional linear model which took $z_\psi$ as input and attempted to predict the true latents $z$. The loss of this linear model was then backpropagated through the context model along with the task loss. Results in Figure 3 (purple) show that this auxiliary loss provided no improvement apart from minor increases in accuracy on Raven's PM, suggesting that incorrect latent variable inference does not explain the explicit model's suboptimal performance. Indeed, when we did not use the auxiliary loss as a training signal for the explicit model and just evaluated the quality of the latent variables learned via linear readouts on $z_\psi$, we found that we could still accurately decode the true latent variable (see Figure 5 (a)). This means that in the absence of any explicit training signal to predict the true latent variable, the explicit model still learns to extract it in order to minimize prediction loss on the query.

Given that the explicit model correctly infers the true latent variable in its bottleneck, we study whether the prediction function is suboptimally learned. In other words, despite the explicit model having access to the true $z$, we hypothesize that $p_\gamma(y_*|x_*, z_\psi)$ does not learn the true data-generating process $y_* = g(x_*, z)$, where $g$ is the true prediction function – e.g., for linear regression $g(x_*, z) = z^T x_*$. To validate this hypothesis, we trained explicit models with the prediction function $g$ hard-coded as an oracle. For instance, in the linear regression task, the $z_\psi$ output by the context model was linearly projected to the same dimensionality as the true weights $z$, after which the prediction function took its dot product with queried input $x_*$. In this setting, if the explicit model extracts the correct latent variable, it should generalize perfectly both in and out of distribution. Our results in Figure 3 confirm that using the correct prediction function indeed provides substantially better OOD generalization and effectively solves most tasks. This finding has significant implications: it suggests that while learning the true task latents may be a necessary condition for generalization, this must also be supplemented with significant inductive biases in the

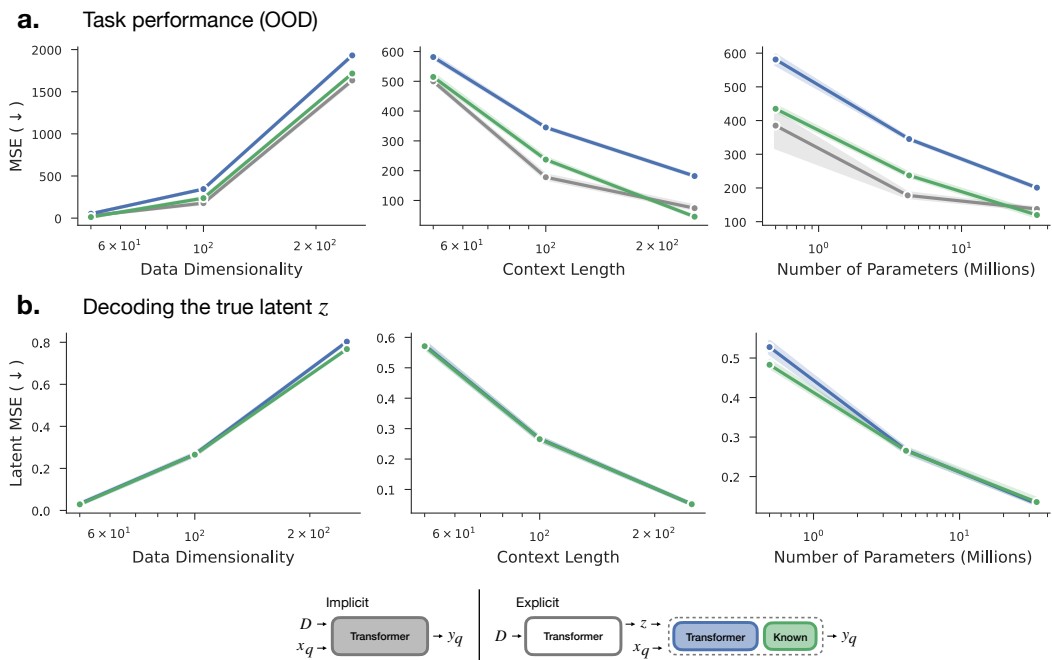

Figure 6: We analyze (a) Linear regression OOD task performance and (b) latent variable linear decoding performance as a function of model and task parameters. Task performance scales similarly for implicit (gray) and explicit models with Transformer prediction (blue). Models in green use the true prediction function $g$. Latent variable decoding accuracy scales as a function of the latent variable's inherit uncertainty (data dimensionality and context length) and the model's size.

prediction function – for instance, through novel architectures – so that these task latents can be leveraged correctly. In the absence of such inductive biases, inferring the correct task latent appears to provide no benefits in practice. We do note that our nonlinear regression tasks, where $z$ represents the weights of an underlying MLP generating the data, were an exception to the results described here in that using an oracle prediction function performed poorly. In this case, we conjecture that the underlying latent variable is too difficult to accurately infer from the context, while shortcut-based solutions would avoid latent variable inference altogether to provide robust solutions.

**Explicit models are easily interpretable.** While explicit models do not provide performance benefits, the ability to extract the correct latent and summarize it in a single bottleneck can still be useful for interpretability. On tasks with a known underlying latent variable, we were able to linearly decode it from $z_\psi$ with high accuracy in most cases, meaning that the information is not only present but also easily accessible (Figure 5 (a)). The exceptions were on complex nonlinear regression tasks where the latents are difficult to infer and classification tasks where more context samples are needed to precisely identify the true decision boundaries. In contrast, finding such clear task-relevant latents is immensely challenging in an ordinary Transformer trained to do ICL, given that they can be distributed across a mixture of many layers and token positions.

Furthermore, even when latent variables appear to be successfully identified using a linear decoder in some hidden layer of a Transformer, one often finds that the relationships merely amount to spurious correlations (Ravichander et al., 2021). For instance, if one manually modifies the activations in this hidden layer such that a different latent variable is predicted by the linear decoder, the model's predictions do not generally change in a way that is "counterfactually" consistent with this new latent (i.e., the prediction is not what it should have been under the new latent variable). We therefore used the Distributed Alignment Search (DAS) method from Geiger et al. 2023b (see Appendix C.4) to search for units in the implicit and explicit models that can be manipulated to obtain correct counterfactual predictions. For the explicit model, we limited this search to the bottleneck $z_\psi$. We found that using the explicit model, we were indeed able to manipulate $z_\psi$ and obtain correct counterfactual predictions, but we were not able to successfully do this using the implicit model, as shown in Figure 5 (b). Explicit models might therefore be useful for both mechanistic interpretability and scientific discovery (Geiger et al., 2023a), where we do not know the underlying task latents

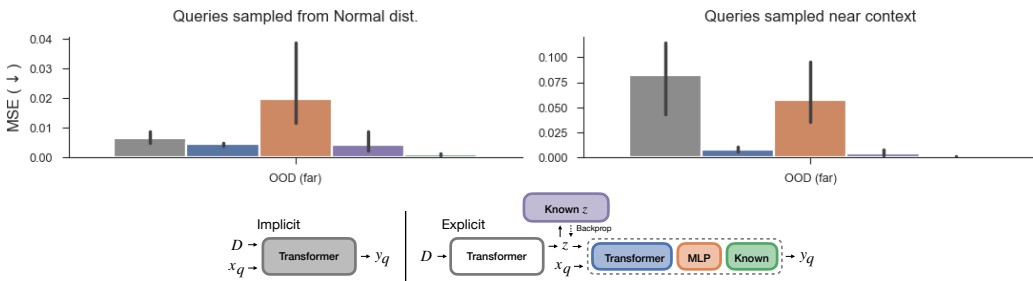

Figure 8: We analyze sinusoid regression where during training, the query is either sampled from the normal distribution (*left*) or close to the context points (*right*). At evaluation, query points are sampled far from the context. Our results indicate that when queries are sampled close to the context, implicit models can rely more on kernel-regression based nearest-neighbor solutions which don't generalize far from context, while explicit models do.

and want to be able to easily uncover them from the trained model, and subsequently obtain a good predictor for an intervened system zero-shot given some knowledge about the intervention.

**Scaling Trends across Different Properties.** To better compare the implicit and explicit models, we investigated their OOD task performance on linear regression as we varied the different properties of the task (input dimensionality and context length) and the size of the model used (Figure 6 (a)). We found that performance in both models scaled similarly, but that the implicit model reliably outperformed the explicit one unless it used the known prediction function $g$. We also looked at the latent variable linear decoding accuracy in the explicit model as a function of these task and model properties (Figure 6 (b)). As expected, we found that the latent variable was easier to decode from the explicit model's bottleneck when there was less inherit uncertainty about its value (lower data dimensionality, longer context length) and when the explicit model was given more capacity. However, throughout the different settings, we see that while the explicit model does learn the true latent well, it is not sufficient to get a performance boost over the implicit models. Further details on the setup of these scaling experiments is provided in Appendix C.2.

**Impact of auxiliary loss on decoding from bottleneck.** Additionally, we perform an experiment where instead of using an auxiliary loss obtained between the ground-truth task latents $z$ and a decoding from the explicit model's bottleneck (called aux_decoded), we instead force the bottleneck itself to be directly close to the ground-truth (called aux_direct). Since the prediction function relies on the bottleneck and not its decoding, removing this extra layer when providing additional supervision might allow the bottleneck to better reflect the task latents and thereby aid prediction. Our results, however, indicate that doing so does not lead to any benefits on OoD evaluation for linear regression, further strengthening the conclusion that effective task latent inference is not the biggest problem in such models.



Figure 7: We analyze the difference between using the auxiliary ground-truth task latent loss directly on the output of context aggregation, i.e. $z_\psi(\mathcal{D})$ which gets fed to the prediction network (aux_direct), or to a linear decoding from it (aux_decoding).

**Extreme Shortcut Injection.** Finally, we test whether injecting some extreme shortcuts during training pushes implicit models to learn nearest-neighbor styled kernel-regression shortcuts as opposed to uncovering the underlying functional form. To study this, we leverage the sinusoid task where we contrast training on queries sampled either arbitrarily or specifically close to the context; with evaluation on query coming far from the context. Our results in Figure 8 indicate that while implicit models perform well generally, they suffer considerably more in the presence of such injected shortcuts since explicit models distill the task latent from context independent of the query.

We further refer the readers to Appendix D for additional analysis into our empirical results.

## 5 CONCLUSION

A commonly believed hypothesis is that Transformers do ICL through brittle statistical shortcuts rather than by inferring the underlying generative latent variables of the task, and that this explains their inability to generalize outside of the training distribution. Here, we empirically tested this hypothesis by minimally modifying the Transformer architecture through the use of a bottleneck that factorized the model into separate context aggregation and prediction functions, creating an inductive bias for explicit latent variable inference. While we confirmed that this model indeed learned to infer the correct latent variables across many ICL tasks, it surprisingly gave no improvement in performance for either in-distribution or out-of-distribution evaluation. Contrary to common belief, then, we showed that simply learning the correct latent variables for the tasks is not sufficient for better generalization because end-to-end optimization does not learn the right prediction model to leverage these latent variables. Indeed, when we substituted the prediction function with a known oracle, we often saw significant advantages in OOD performance. Importantly, we were able to make such findings because we performed controlled experiments on simple tasks with known latent variables and prediction functions, thus avoiding the complexities and confounding variables that come with natural language and pre-trained LLMs. Our results point to a line of future work that incorporates inductive biases in the prediction model to better leverage the inferred latent variables, and motivates current efforts in improving (a) amortized methods for in-context prediction (e.g., Garnelo et al., 2018a;b; Kim et al., 2019; Hollmann et al., 2022) and (b) neurosymbolic AI which tries to combine flexible LLMs for inference with highly structured functions for prediction (e.g., Wong et al., 2023; Mialon et al., 2023; Ellis, 2023).

**Limitations.** While our work investigates potential reasons behind the shortcomings of current ICL approaches, overcoming those shortcomings is an entirely different and bigger problem which is a relevant future direction but not an achievement of the current work. We also look at ICL solely from the perspective of algorithmic tasks with models trained from scratch, without exploring in detail the connections with ICL in pre-trained models.

**Broader Impact.** Our approach paves the way into an interpretable model for in-context learning, which could prevent harmful use of such models by additionally encoding constraints on the interpretable bottleneck. Additionally, our work provides a better understanding about ICL which can lead to safer future models.

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

## A  RELATED WORK

**In-Context Learning.** In-Context learning (ICL) is an ability of certain trained models to take an entire task's dataset as input and parameterize solutions directly in their layer activations, which then condition subsequent computation on novel inputs from those same tasks. Generally, this ability is found in sequence models such as Transformers where the task dataset, or "context", corresponds to an earlier part of the sequence. ICL was first observed in large-scale pre-trained LLMs (Brown et al., 2020), and is similar in many respects to meta-learning (Chan et al., 2022). These LLM findings were subsequently expanded to more controlled settings outside of the language modality, where Transformer models were directly trained on task distributions such as linear regression (Von Oswald et al., 2023; Garg et al., 2023), hidden Markov models (Xie et al., 2022), compositional grammars (Hahn & Goyal, 2023), regular languages (Akyürek et al., 2024) and Turing machines (Grau-Moya et al., 2024), with a set of task observations defining the "context". These works highlight that Transformers are indeed able to model many types of complex task distributions, approaching in many cases the performance of the Bayes-optimal predictor, the a-priori optimal solution (Xie et al., 2022). Our work lies along similar lines but using more complex tasks and a systematic study into the differences between modeling the predictive space directly, or through a two-step process involving explicit inference of task latents.

**Shortcuts in ICL.** Shortcut learning is a phenomenon that has widely been observed in machine learning (Geirhos et al., 2020), and refers to where a model solves a task through statistical correlations that are accidental and thus not robust to even slight distribution shifts. A classical example of this in image classification is the usage of background cues to classify objects (Ribeiro et al., 2016). Similar mistakes are know to be very common in NLP (McCoy et al., 2019), specifically in reasoning tasks (Zhang et al., 2022). Particularly relevant to our work, many authors have shown that has shown that Transformers are very prone to relying on shortcuts when performing ICL Tang et al. (2023). For instance, Olsson et al. (2022); Singh et al. (2024) have shown that *induction heads* play in important role in ICL by predicting that the continuation of a token will be the same as last time (i.e. $[a][b] \ldots [a] \rightarrow [b]$). As shown by Von Oswald et al. (2023) this motif can be used to do linear regression, and can generally be seen as a form of kernel regression (i.e. $p(y_q|x_q, x_{1:n}) \propto \sum_i K(x_i, x_q)y_i$, (Han et al., 2023)). This observation draws a link between those types of solutions and non-parametric inference methods in statistics (Hastie, 2009), of which kernel regression is a member. In contrast (Hendel et al., 2023; Todd et al., 2023) have concurrently shown that in some cases Transformers encode a "task vector" that they infer from the context and then use to do the prediction. There is therefore a need to better understand the nature of shortcuts in ICL and whether or not they can be easily avoided for better generalization. Our work explores this very question.

**Neural Processes.** The problem of solving new tasks in a zero-shot manner directly at inference is also closely tied to amortized Bayesian models (Kingma & Welling, 2013; Rezende et al., 2014; Radev et al., 2020; Geffner et al., 2023; Mittal et al., 2023). Conditional Neural Processes (CNPs) (Garnelo et al., 2018a) provide a framework akin to the explicit model, where the posterior predictive distribution is modeled through a bottleneck $z_\psi$, i.e. $p_\theta(y_*|x_*, \mathcal{D}) = p_\theta(y_*|x_*, z_\psi(\mathcal{D}))$. However, CNPs do not look at the relevance of $z_\psi$ to the true latent $z$, and use the DeepSets (Zaheer et al., 2017) architecture to model $z_\psi$, though recent research generalizes this setting to use Transformers Nguyen & Grover (2023) and other architectural backbones as well (Kim et al., 2019; Gordon et al., 2019). Our approach with the explicit model, however, is to precisely question whether task-specific latents are encoded via $z_\psi$ which is now instead modeled using a Transformer architecture. Analogously, Neural Processes (NPs) (Garnelo et al., 2018b; Pakman et al., 2020) augment CNPs with probabilistic modeling, where $z$ is now modeled explicitly as a latent-variable in the Bayesian sense, i.e. the likelihood is now modeled as $p_\theta(y|x_*, \mathcal{D}) = \int p_\theta(y|x, z)p_\theta(z|\mathcal{D})dz$, where $z$ represents the latent variable and $\theta$ the parameters of the likelihood model. The model is trained via the Evidence Lower-Bound (ELBO) with the amortized variational approximation $q_\varphi(\cdot|\mathcal{D})$. Once trained, predictions for new datasets can be made by simply performing inference over the *encoder* $q_\varphi$ to obtain $z$, and then leveraging this latent variable to eventually give the predictions via $p_\theta(y|x_*, z)$. Hence, while CNPs and the explicit model to share similarities in architecture, our goal is orthogonal in that we specifically use the explicit model to understand the impact of task-specific latent variable inference on ICL setups.

**Meta-Learning.** Meta-learning (Hospedales et al., 2020b;a) studies systems that can learn over two levels: rapidly through an inner-loop that is meta-learned using a slower outer-loop. The goal in such methods is to learn a good initialization common to the parameterized family of tasks, in

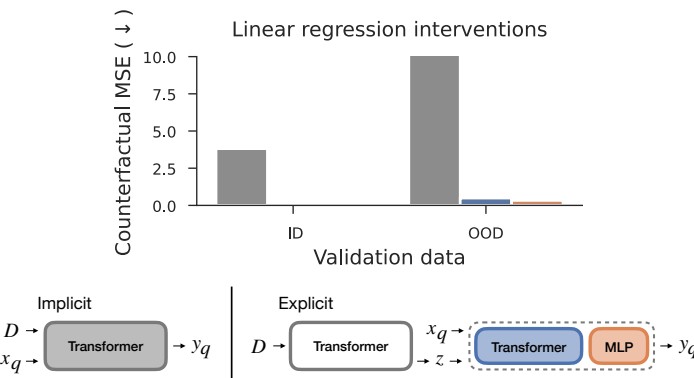

Figure 9: Same experiment as Figure 5b but with the linear regression task. Specifically, we use Distributed Alignment Search (DAS, Geiger et al. (2023b), see Appendix C for details) to find the 10 dimensional subspace in each model with best simulates counterfactual interventions on the task vector (in this case, the weight of the linear regression). In both explicit model, the subspace is taken at the bottleneck. In the implicit one, we perform the DAS at all layer of the query token and report the best one. The reported metric is the MSE of the intervened model on the intervened regression problem (i.e. using the same query $x$ but $y$'s coming from the intervened linear regression weights, Appendix C for details).

a manner that obtaining a particular solution for a new task is fast from this initial point. The inner loop provides an optimization trajectory for a randomly drawn task from some initialization, which in itself is optimized in the outer loop to a good solution applicable for the global set of tasks. Typically, evaluation is done on some meta-validation set of tasks not seen during training. Task distributions can for example be a set of different classification/regression tasks (few shot learning, Vettoruzzo et al. (2023)) or variations of a reinforcement learning (meta-RL, Beck et al. (2023)). The goal is similar to ICL approaches in the sense that given a novel context $\mathcal{D}$, one wants to make predictions for some query $x_*$. However, a big difference is that ICL approaches bypass modeling a common initialization by working directly on the prediction side (implicit), or instead predict the optimal parameters directly zero-shot through inference on the context model (explicit).

**Mechanistic Interpretability.** Mechanistic interpretability is interested in understanding deep neural network's computations through interpretable abstraction, akin to what computational neuroscience does with the brain. Alain & Bengio (2018) introduced the foundational technique of linear "probes", which are linear models trained on the hidden state of a network to predict an abstract feature of the input; the success of which suggests that such a feature is used by the model. Since then, this naïve approach has been criticized for being potentially misleading (Ravichander et al., 2021); in many cases a feature can be linearly decoded from a model without the model using it. More reliable methods grounded in causality (Vig et al., 2020; Geiger et al., 2024) have now became the gold standard, and their use applied to Transformers has been exploding in popularity (Elhage et al., 2021).

# B TASKS

We consider the following tasks for our evaluations, specified by the data-generating prediction function $g : x, z \rightarrow y$ which is used to generate the ICL dataset, where $z$ represents the task-specific latent variable.

## B.1 REGRESSION TASKS

For regression tasks, since $y \in \mathbb{R}$, we use the mean-squared-error loss to train the model.

**Linear Regression.** This refers to the task where $y$ is obtained from an affine transformation on the input $x$. In particular, $y = g(x, z) = z^T x$, where $z \in \mathbb{R}^{i \times j} \sim \mathcal{N}(0, I)$. For our experiments, we set $\dim(x) = 1$ and $\dim(y) = 1$.

**Nonlinear Regression using MLPs.** Here, the labels $y$ are obtained from a neural network which takes $\boldsymbol{x}$ as an input. In particular, $\boldsymbol{y} = g(\boldsymbol{x}, \boldsymbol{z}) = f_{\boldsymbol{z}}(\boldsymbol{x})$, where $f_{\boldsymbol{z}}$ is modeled as a Multi-Layer Perceptron (MLP) network with a $64$ dimensional single hidden layer and ReLU nonlinearity. The distribution of the weights of the neural network is $\boldsymbol{z} \sim \mathcal{N}(0, I)$. For our experiments, we set $\dim(\boldsymbol{x}) = 2$ and $\dim(\boldsymbol{y}) = 1$.

**Sinusoid Regression.** For this task, the label $y$ is obtained as a summation of sine functions with different frequencies and amplitudes, taking $\boldsymbol{x}$ as an input. Mathematically, we structure the system as $y = g(\boldsymbol{x}, \boldsymbol{z}) = \sum_{i=1}^{K} \alpha_i \sin(2\pi\lambda_i \boldsymbol{x})$, where $\lambda_i$'s denote the frequencies and $\alpha_i$'s the amplitudes. The parameters for the system can be seen as $\boldsymbol{z} = \{\alpha_{1:K}\}$ while $\lambda_{1:K}$ remains fixed throughout. Additionally, for our experiments we set $K = 3$, and consider the distributions $-\lambda_i \sim \mathcal{U}(0, 5)$ and $\alpha_i \sim \mathcal{U}(-1, 1)$, and set $\dim(\boldsymbol{x}) = 2$ and $\dim(y) = 1$.

**Gaussian Process Regression.** While the other tasks considered had a parametric nature to it, this task on the other hand has more of a non-parametric nature. Here, the task is that $\boldsymbol{Y} \sim \mathcal{N}(\boldsymbol{0}, K(\boldsymbol{X}, \boldsymbol{X}))$, i.e. the set of labels is sampled from a joint Gaussian distribution, akin to drawing a random function through a Gaussian Process (GP) prior and then evaluating it at different points $\boldsymbol{X}$; with $K$ defining the kernel in the GP. In our case, we consider $K(\boldsymbol{x}, \boldsymbol{x}') = \exp\left(-\frac{\|\boldsymbol{x}-\boldsymbol{x}'\|^2}{2\sigma^2}\right)$ as the RBF kernel and $\boldsymbol{X} = (\boldsymbol{X}_c, \boldsymbol{X}_q), \boldsymbol{Y} = (\boldsymbol{Y}_c, \boldsymbol{Y}_q)$ are the combined points for both the context and the queries, which are split after this sampling. Here the latents $\boldsymbol{z}$ has to store the kernel computations between the query and all the context points $\boldsymbol{X}_c$, as well as the corresponding context labels $\boldsymbol{Y}_c$. Storing this either involves storing the high-dimensional mapping of $\boldsymbol{X}_c$ which is defined by the kernel $K$, or storing all the points $\boldsymbol{X}_c$ themselves. This is thus very high dimensional and weakly structured.

**Hodgkin-Hoxley ODE Prediction.** This is an example of the task where the context $\mathcal{D}$ is not composed of *iid* entries, but instead observations from the Hodgkin-Huxley temporal dynamics model of neural activity unrolled through time :

$$C_m \frac{dV}{dt} = g_1(E_1 - V) + \bar{g}_{Na}m^3 h(E_{Na} - V) + \bar{g}_K n^4(E_K - V) + \bar{g}_M p(E_K - V) + I_{inj} + \sigma\eta(t)$$

Above, $V$ represents the membrane potential which is the target of interest, $\boldsymbol{t}$ represents the different points at which observations are provided, $C_m$ is the membrane capacitance, $g_l$ is the leak conductance, $E_l$ is the membrane reversal potential, $\bar{g}_c$ is the density of channels of type $c$ (Na$^+$, K$^+$, M), $E_c$ is the reversal potential of $c$, $(m, h, n, p)$ are the respective channel gating kinetic variables, and $\sigma\eta(t)$ is the intrinsic neural noise. The right hand side of the voltage dynamics is composed of a leak current, a voltage-dependent Na$^+$ current, a delayed-rectifier K$^+$ current, a slow voltage-dependent K$^+$ current responsible for spike-frequency adaptation, and an injected current $I_{inj}$. Channel gating variables $q$ have dynamics fully characterized by the neuron membrane potential $V$, given the respective steady-state $q_\infty(V)$ and time constant $\tau_q(V)$ (details in Pospischil et al. (2008)).

Importantly, in our experiments, we fix all parameters but $(\bar{g}_{Na}, \bar{g}_K)$ to values in Tejero-Cantero et al. 2020 and solve the differential equation for 6,400 pairs $(\bar{g}_{Na}, \bar{g}_K) \in [0, 40]^2$ from $t = 0$ to $t = 120$ with 1000 time-steps. In other words, the Transformer has to regress to solutions of ordinary differential equations, where the task latents are $\boldsymbol{z} = \{\bar{g}_{Na}, \bar{g}_K]\}$, the observations are $\boldsymbol{x} = \boldsymbol{t}$ and $y = V$, such that $y = g(\boldsymbol{x}, \boldsymbol{z})$. Here $g$ represents the unrolling of the differential equation.

### B.2 CLASSIFICATION TASKS

For classification tasks, since $y$ is a categorical measure, we use a cross-entropy loss for training.

**Linear Classification.** Akin to linear regression, here we consider the case that $y$ is obtained by an affine transformation of $\boldsymbol{x}$ followed by a sigmoid function and a consequent sampling step. That is, $y = g(\boldsymbol{x}, \boldsymbol{z}) \sim \text{Categorical}(\text{Softmax}(\boldsymbol{z}^T \boldsymbol{x}))$ where $\boldsymbol{z} \in \mathbb{R}^{i \times j} \sim \mathcal{N}(0, I)$. For our experiments, we set $\dim(\boldsymbol{x}) = 2$ and $y \in \{0, 1\}$.

**Nonlinear Classification Using MLPs.** Here, the logits for the labels are instead obtained through a neural network taking $\boldsymbol{x}$ as an input, and not an affine transformation. Mathematically, this can be seen as $y = g(\boldsymbol{x}, \boldsymbol{z}) \sim \text{Categorical}(\text{Softmax}(f_{\boldsymbol{z}}(\boldsymbol{x})))$ where $f_{\boldsymbol{z}}$ is modeled as a Multi-Layer Perceptron (MLP) network with a $64$ dimensional single hidden layer and ReLU nonlinearity. The

distribution of the weights of the neural network is $\boldsymbol{z} \sim \mathcal{N}(0, I)$. For our experiments, we set $\dim(\boldsymbol{x}) = 2$ and $y \in \{0, 1\}$.

### B.3 COMPOSITIONAL TASKS

**Reusable Modular Mixture of Experts (MoE).** We consider a modular task which consists of sequential application of a choice of $K$ experts $g_1, \ldots g_K$ over the input $\boldsymbol{x}$. In particular, the computational graph consists of $L$ layers where at each layer $l$, expert $\boldsymbol{z}_l$ operates on the output of the preceding layer to give the successive output, i.e. $\boldsymbol{x}^l = g_{\boldsymbol{z}_l}(\boldsymbol{x}^{l-1})$. This task is compositional in nature because at each layer, any of the $K$ experts can be called upon to perform a unit of computation and the choice of the expert is defined by the underlying task latent $\boldsymbol{z}_1, \ldots, \boldsymbol{z}_L$, each of which are categorical with $K$ possibilities. In our specific implementation, we set $L$ to be 5, $K$ to be 5, $\boldsymbol{x} \in \mathbb{R}^4$ and $\boldsymbol{y} \in \mathbb{R}^4$. Each expert $g_i$ is parameterized as a linear layer followed by the tanh activation function. We enumerate all $K^L$ possible combinations and then only use a subset of them during training, while randomly sampling all for evaluation.

**Alchemy.** Alchemy is a meta-reinforcement learning benchmark Wang et al. (2021) where each environment is defined by a set $\mathbf{z} = (\text{GRAPH}, \text{POTION MAP}, \text{STONE MAP})$ of rules about how some set of potions transforms some stones. We extracted from it an ICL classification dataset consisting of transformations $\mathbf{x} = (\text{STONE}, \text{POTION}) \rightarrow \mathbf{y} = \text{STONE}$. The transformations are compositional and symbolic; each potion affects only one of the three properties of stones (size, shape and color). An environment is specified by how observable stones and potions MAP to latent stones and potions, along with a GRAPH over these latent stones which specify the result of the Transformations. In total there is 109 GRAPH, 48 POTION MAP and 32 STONE MAPS, making for 167424 environments. We reserve 100,000 environments for evaluation and train of the remaining ones.

**Raven's Progressive Matrices.** Raven's Progressive Matrices (Raven's PM) is a reasoning task used for IQ tests (John & Raven, 2003). It consists of a 3x3 grid where each cell contains simple objects varying in a small number of attributes (number, shape, size, color), but the bottom right cell is left empty. Subjects must notice a pattern in how the cells change from left to right in the first two rows of the grid, and then use that same pattern to complete missing cell in the bottom row. This is done by selecting one answer among $N$ possible provided options for the missing cell. We use a symbolic version of the dataset that addresses bias in the original version (Guo et al., 2023). In this dataset, objects at a cell have 4 discrete attributes with 40 possible values each. In our models, the context consists of the first two rows of the grid, the query consists of the last row with a masked out final cell, and the ground-truth latent variable is the underlying rule that generates a particular grid. Each rule is composed of a set of sub-parts, and we evaluate on unseen compositions.

**Gene Targeting.** We use Perturb-seq dataset collected by Norman et al. (2019) where researchers performed several genetic intervention experiments using CRISPR (Gilbert et al., 2014). In each experiment, either one or two genes were targeted and the resulting expressions across 5000 genes were observed across several cells. Here, we consider each CRISPR intervention experiment as a different context, the resulting cell genetic expressions as 5000-dimensional observations, and a left-out cell with half of the genetic expressions randomly masked out as the query. The task is to predict the missing genetic expressions for the queried cell. We evaluate on held on held out CRISPR experiments with novel pairs of targeted genes.

## C MODEL DETAILS

In the following section, we describe the standard architectural details used for all the tasks, as well as specific differences in the architecture used for the scaling experiments. Finally, we also provide details about the distributed alignment search mechanism.

### C.1 GENERAL DETAILS

For our implicit model, we use a standard Transformer with 8 layers. In the explicit model, for context aggregation we parameterize $\boldsymbol{z}_\psi(\mathcal{D})$ using a standard Transformer with 4 layers, 256 dimensions latent, 512 dimensions MLP, and 4 heads. For the predictor $p_\gamma$, we consider two options:

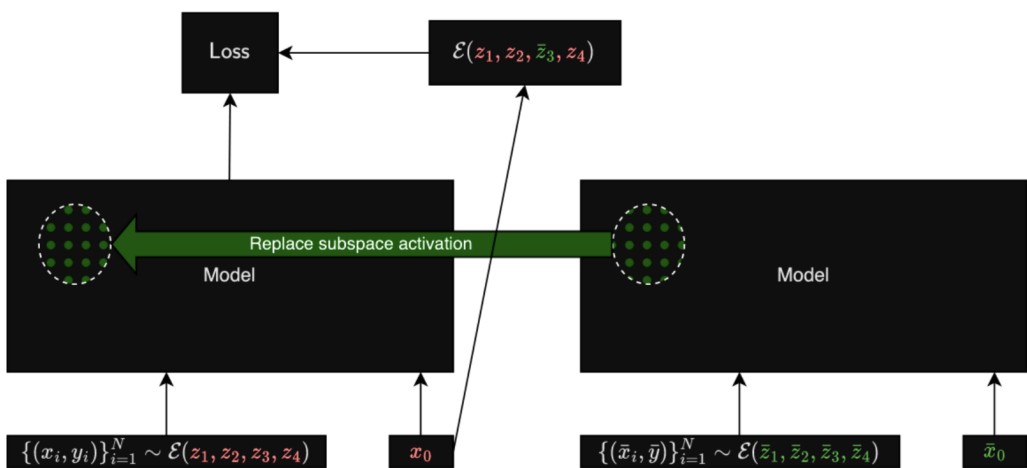

Figure 10: Illustration of the DAS training procedure

a ReLU-actiavtion based MLP with three hidden layers of size 512 and a Transformer with the same configuration as $\boldsymbol{z}_\psi(\mathcal{D})$.

For the implicit model, we format the prompt for prediction as $[\boldsymbol{x}_1, y_1] \ldots [\boldsymbol{x}_n, y_n][\boldsymbol{x}_q, \emptyset]$, where every $[\cdot]$ represents a token. We use a distinct mask token $\emptyset$ to represent the target (which is the thing being predicted). For the explicit model, we first compute $[\boldsymbol{x}_1, y_1] \ldots [\boldsymbol{x}_n, y_n]$ to $\boldsymbol{z}_\psi(\mathcal{D})$ with the context Transformer, then we give $[\boldsymbol{z}_\psi(\mathcal{D})][\boldsymbol{x}_q]$ to the predictor Transformer or $[\boldsymbol{z}_\psi(\mathcal{D}), \boldsymbol{x}_q]$ to the MLP.

For our experiments, the number of context points $n$ is uniformly sampled from 16 to 128 for both training and evaluation. Training is done with new data being synthetically generated on the fly, and evaluation either based on the test set provided for real-world tasks or simulated data of 1000 different contexts for synthetic tasks. All the models were trained with a learning rate of $10^{-4}$ using the Adam optimizer (Kingma & Ba, 2014) for a 1000 epochs.

### C.2 SCALING EXPERIMENTS

For the scaling experiments, we only consider the linear regression case with a base configuration of: (a) $\boldsymbol{x}$ of dimensionality 100, (b) context size being sampled uniformly from $(75, 125)$, and (c) 8 heads, 8 layers, 512 hidden dimensions and 256 bottleneck dimension for the transformer models.

From this base configuration, we changed only one of the configurations at each time to test for scaling trends for each property independently. In particular, we ablated over $(50, 100, 250)$ for the dimensionality of $\boldsymbol{x}$, $(50, 100, 250)$ for the context length which was sampled from a $\pm 25$ range and the model size. The smallest model size considered had 4 heads, 4 layers, 256 hidden dimensions and 128 feature dimensions. The medium multiplied each of these properties by $2\times$, and the biggest model subsequently multiplied it by $2\times$ again. For the explicit models, we considered the same scaling paradigms with the number of layers being split by half to accommodate a separate context model and prediction model.

All the models were trained with a learning rate of $10^{-5}$ using the Adam optimizer for 5000 epochs.

### C.3 COMPUTE DETAILS

We train most of our models on single RTX8000 NVIDIA GPUs, where it takes roughly 3-6 hours for each experiment to run. Our scaling experiments on the other hand often required 1-2 days on single GPUs for training each model.

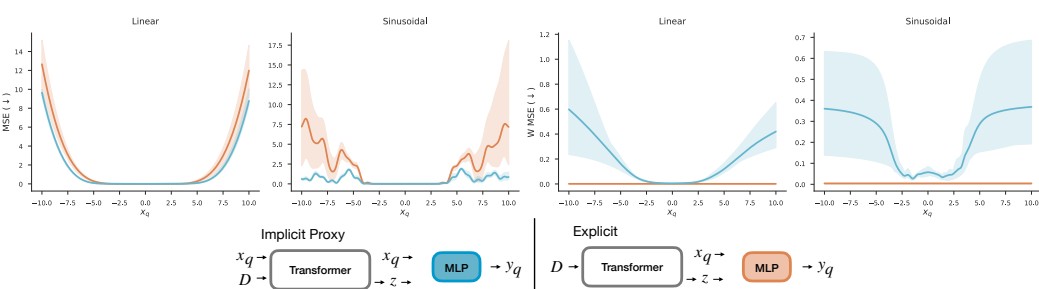

Figure 11: To further understand the difference between the explicit and implicit model, we utilize an implicit proxy model which shares the same architecture as the explicit model with MLP prediction with just *one key difference*: the task latent $z$ depends on the query $x_q$ as well. This task latent $z$ can be understood as the final attention layer output of the implicit model, after which an additional MLP is utilized to provide prediction. Our findings on linear and sinusoidal regression demonstrate that as we move further and further out-of-distribution, the implicit proxy model performs better than the explicit model (left figures), but recovers the underlying true task latents worse (two right figures). This provides additional validation of our hypothesis.

## C.4 DISTRIBUTED ALIGNMENT SEARCH DETAILS

To find subspace causally associated with a task latent in Alchemy, we use a method based on Distributed Alignment Search (DAS) by Geiger et al. (2023b). This procedure is performed for a location $L = \mathbb{R}^d$ (e.g. the bottleneck) and latent $i \in \{1, 2, 3\}$ (GRAPH, STONE MAP, POTION MAP).

First, we run with the model on $\mathcal{D}_z$ and $\mathcal{D}_{\bar{z}}$ for every possible query $x_*$. We call $z$ the base and $\bar{z}$ the source and only differ by the $i$th latent. For every run, we record the activity of the source model at the location $l_z \in \bar{L}$. Then, we run the base model again but this time fixing the subspace of $l$ defined by the orthogonal projection $\Pi \in \mathbb{R}^{d \times 10}$ to it's value in $l_z$. A single projection $\Pi$ is learned over all possible combination $z, \bar{z}$ and $x_*$ with a cross-entropy loss between the prediction of the base (intervened) model and the true counter-factual result of changing the latent $z_i$ to $\bar{z}_i$. See Figure 10 for an illustration of the process. A subspace is evaluated by looking at the accuracy of the counterfactual interventions over a dataset of held-out $z, \bar{z}$ pairs; a quantity called the Interchange Intervention Accuracy (IIA). In Figure Figure 5 (b) we report the validation IIA relative to a baseline corresponding to the counterfactual accuracy if we don't perform any intervention (because changing the latent sometimes doesn't change the prediction) $\frac{\text{IIA} - \text{BASELINE}}{1 - \text{BASELINE}}$.

## D ANALYSIS OF EXPERIMENTS

Based on the empirical evidence presented in Section 4, we finally provide details and analysis into the results to further the understanding of the conclusions. In particular, our key analysis includes

**Explicit Models sufficiently uncover task latents.** We see that in problems where the context provides enough evidence to uncover the true task latents, explicit models are able to do so. In particular, this hints at the fact that explicit models do perform downstream prediction based on true task latents whenever these latents can be sufficiently identified from the context examples.

**Explicit Models do not generalize better than implicit ones.** Our analysis also reveals that while explicit models often do uncover the right task latents, they are still not able to surpass implicit models even on OOD generalization. This could be due to implicit models also uncovering the true underlying prediction function but in a distributed fashion, or explicit models not being able to leverage the learned latents in downstream prediction.

**Learned downstream prediction is often sub-optimal.** Our results indicate that it is indeed the case that while the explicit models do uncover the right latents, they fail to generalize well OOD because the downstream prediction function fails to generalize.

**Classification tasks vs. regression tasks.** OOD performance is generally strong (across all models) for classification because decision boundaries are within the training domain and do not change

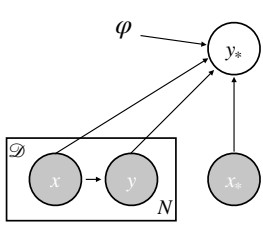 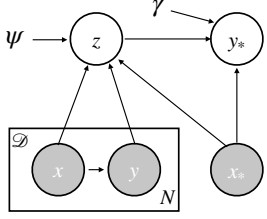 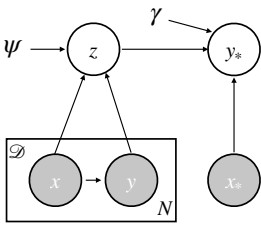

Figure 12: Plate diagram for the implicit model (left), implicit proxy model (middle) and the explicit model (right), where gray blocks refer to observed variables and white refers to unobserved variables. Trainable parameters are indicated without circles. In the explicit model case, $z$ is currently modeled as a dirac measure defined via the trainable parameters $\psi$. One can see the implicit proxy model as very similar to the implicit model where output of the last attention layer corresponding to the query token is further processed to give prediction. Its similarity to the explicit model is also clear as it shares exactly the same parameterization.

beyond it. In contrast, for regression tasks, the function continues to change beyond the observed training domain, making OOD prediction more difficult. This is also why known prediction functions give little benefit in classification tasks: they are already solved well OOD with ordinary implicit and explicit models.

**The explicit model with known prediction function does not give benefits in nonlinear (MLP) regression.** This is because the problem of inferring an MLP's weights given some context examples is too difficult, so the explicit model opts for a different, non-parametric solution. This is supported by the latent variable decoding results in Fig. 5 (previously Fig. 4), which show that even with a known prediction function the explicit model does not learn to infer the correct latent variable for the nonlinear (MLP) regression task.

# E   MATHEMATICAL FORMALISM

In this section, we provide a formal distinction between the implicit and explicit model. In both the approaches, the goal is to model the true posterior predictive $p(y|\boldsymbol{x}, \mathcal{D})$; however the two methods model it through different conditional independence setup.

**Implicit Model**. In this setup, we model the predictive distribution as $p_\varphi(y|\boldsymbol{x}, \mathcal{D})$, where the training is done as

$$\arg\max_{\varphi} \mathbb{E}_{\boldsymbol{x}, y, \mathcal{D}} \left[\log p_\varphi(y|\boldsymbol{x}, \mathcal{D})\right] \tag{2}$$

and then given a query $\boldsymbol{x}$ and dataset $\mathcal{D}$, the inference is done simply by sampling or estimating the mean of $p(y|\boldsymbol{x}, \mathcal{D})$.

**Explicit Model**. Contrary to the implicit model, the explicit model parameterizes the predictive distribution as $p_\gamma(y|\boldsymbol{x}, \boldsymbol{z}_\psi(\mathcal{D}))$, with a similar training procedure as above. Note that the predictive distribution only interacts with the dataset $\mathcal{D}$ through the latent $\boldsymbol{z}_\psi(\mathcal{D})$ while the implicit model allows unconstrained access to $\mathcal{D}$.

**Implicit Proxy Model**. To better understand the differences that play a role from architectural differences and parameterizations, we use exactly the same architecture as the explicit model to obtain a version of the implicit model. Such a model parameterizes the predictive distribution as $p_\gamma(y|\boldsymbol{x}, \boldsymbol{z}_\psi(\mathcal{D}, \boldsymbol{x}))$, with a similar training procedure as above. Note that the only difference with the explicit model here is that the conditional dependence of the query and the task latents is broken.

We refer the reader to Figure 12 for a plate diagram of the corresponding architectures.

