# OpenReview forum: "Does learning the right latent variables necessarily improve in-context learning?"
_ICLR.cc/2025/Conference — Submitted to ICLR 2025_

### Official Review · Reviewer_q3Zg · 2024-10-24

**Soundness:** 2
**Presentation:** 3
**Contribution:** 2
**Rating:** 5
**Confidence:** 3

**Summary:**

This paper investigates the role that learning ground-truth task vectors plays for out-of-domain generalization in the in-context learning framework. For this purpose, the authors compare a traditional transformer (referred to as the “implicit model”) and a similar model with a bottleneck in its architecture (the “explicit model”).
They empirically find that the implicit model does not perform better than the explicit model, on the OOD setup, even though the implicit model correctly learns a relatively good representation of the ground-truth task vector. They thus demonstrate that access to the task vector is not sufficient for OOD generalization in the ICL framework.

**Strengths:**

Writing: The paper is accessible and easy to read, which, however, is also facilitated due to the nature of the content. The overall structure is straightforward and easy to follow.
The extensive use of bar-plots makes it convenient to understand the experimental results.

Significance: The question whether ICL can be accurately described via task vectors is certainly interesting and important. Clearly stating this question is thus a central contribution of this paper. I also appreciate that the authors present a negative result. The paper also presents a new experimental setup, compared to for instante that in the paper “In-Context Learning Creates Task Vectors” by Hendel et al.

**Weaknesses:**

In summary, I mainly miss a theoretical framework or at least some mathematical justifications for the things you claim and investigate. Without them a lot of your arguments felt only heuristically motivated and shallow.

First, there are no theoretical results referenced or provided that support the premise that the constrained architecture should generalize better in the first place. The paper would benefit a lot from including mathematical reasoning in that direction. Given that the premise that learning an explicit representation of the task vector is necessary (not sufficient, that’s clear) for generalization is not well-supported in the first place, the results from Figure 2, which refute this premise, cannot be seen as an interesting contribution.

You should also be careful with the general statement that using parametric assumptions inevitably leads to better generalization than relying on non-parametric ones. It is, for instance, well-known that the posterior predictive of a Bayesian Linear Regression model is in fact equal to the posterior predictive of a Gaussian Process with an appropriate covariance function. The paper should thus state much more clearly, what is meant by “the model takes shortcuts”.

Furthermore, the results displayed in figure 4 do not provide as much evidence as claimed in the text for the hypothesis that the model accurately learns the task vector since the discrepancy compared to your trivial baseline is, except for the linear and sinusoidal case, not that large and, furthermore, using the ground-truth decoding function does not yield great benefits except for the linear and sinusoidal cases.

There is further little evidence that the explicit model does not also utilize task vectors in some way or another as your counterfactual approach targets interpretability, which is, however, a different question. On the contrary, the results from “In-Context Learning Creates Task Vectors” are still very valid evidence that “implicit” transformers do in fact use task vectors.

There are further several practical issues with the experiments:
1.) The number of tasks you consider is quite small
2.) The dimensionality of the data-points, which you set to 1 or 2  (see Appendix) is also very restricted
3.) You only consider a single and quite limited way of generating OOD samples for each task.

Finally, some sections of the paper are a bit lengthy or perhaps even unnecessary in the main body of the paper, such as the results displayed in figure 5, or the section on Meta-Learning in the related-works part.

I also have a few issues with the reproducibility of your results:
1.) The code is not provided
2.) Please explicitly mention how many parameters your models have and on how many samples they are trained
3.) You should at least mention what the error bars in your plots represent, i.e. the standard error or the IQR or something else?

There are also a few typos, grammatical issues, and notational errors in the manuscript. For example when denoting functions or in the text on the axes of the plots.

Finally, using a blogpost or a Wikipedia Article as a reference might be a possibility to give the reader a pointer to an accessible source of information, but it should clearly not be used as a sole reference.

As a small detail, I also find it a bit unusual that you neither use weight decay nor a learning rate schedule in your training.

**Questions:**

What is the evidence to believe in the first place that learning task vectors might improve generalization? And, especially, what is the evidence that not learning it leads to poor generalization?

You should also be more careful with the claim that parametric models generalize better than non-parametric ones in OOD scenarios, or, alternatively, also support this claim.

The paper would clearly benefit from providing more empirical evidence for the fact that the task vectors are actually always learned well by the explicit model.

Please also provide more evidence that the explicit model learns the task vector *better than* the implicit one.

The scale of the experiments could be improved in several dimensions (please refer to the "weakness" section)

I also think that the paper would benefit from carefully reading it again to fix typos and other small issues and from shortening it to the recommended length of nine pages.

Please also take the reproducibility issues from above into consideration, with releasing the code being the most important point.

---

> ### Author Response · Authors · 2024-11-21
> **Author Response**
>
> We thank the reviewer for their valuable comments and appreciate that they found that our work tackles an “interesting and important” question. We provide further clarifications and evidence below and hope that it alleviates the concerns raised by the reviewer.
>
> > **…a theoretical framework or at least some mathematical justifications…**
>
> We refer the reviewer to works [3-6] which highlight that the use of an information bottleneck does lead to better generalization. The bottleneck which learns the true underlying latents can be seen as an information bottleneck since it is a low-dimensional embedding of the whole dataset. In particular, the explicit model could solve tasks by either distilling the information from context examples into the bottleneck (i.e. approximating the true latent) or by copying all the context examples in the bottleneck. The latter is disincentivized due to the lower dimensionality of the bottleneck; and thus we see that solely through maximum likelihood based training we are able to uncover the right underlying latents. Thus, based on the above reasoning, we stress that our arguments are not just heuristically motivated.
>
> Further, one can see that the functions represented by the explicit model can actually be represented by the implicit model too; in particular the implicit model could first uncover the right latents in the query token’s representation and then use it for downstream prediction, e.g in [7]. However, with increasing context sizes it is less and less the case that an implicit model’s hypothesis space can be represented by the explicit model. Hence we can argue from the perspective that explicit models trim the potential hypothesis space and thus could enjoy better generalization, as is studied in multiple works on generalization bounds [8].
>
> > **…careful with the general statement that using parametric assumptions inevitably leads to better generalization…**
>
> The reviewer has a valid point and we acknowledge that we weren’t completely clear about it in our draft. We will revise our writing to reflect that the kind of shortcuts we refer to are more kernel-regression / nearest neighbor style shortcuts, especially even when there is an underlying true parametric form for the solution.
>
> It is true that for linear regression the ordinary least squares solution can be written as $w = (X^TX)^{-1}X^Ty$ which can equivalently be seen as both a parametric as well as a non-parametric model through the lens of prediction, i.e. $w^T x_* = y^TX(X^TX)^{-1} x_* = \sum_{i=1}^N k(x_i, x_*) y_i$ where $k(x_i, x_*) = x_i^T(X^TX)^{-1}x_*$. See section A.8 of [1] for a related discussion. However, while this equivalence exists for linear regression, it may not necessarily hold for other, more complex modeling approaches.
>
> We refer the reviewer to the general response where we show that indeed by injecting shortcuts in an extreme fashion, the explicit model which relies less on kernel regression because of the bottleneck ends up generalizing better than the implicit one.
>
> >  **The paper would clearly benefit from providing more empirical evidence for the fact that the task vectors are actually always learned well by the explicit model.**
>
> We perform linear decoding and causal interventions to test for whether the explicit model does indeed infer the task vectors correctly. The only cases where we see the explicit model not being very successful at obtaining task latents is in nonlinear MLP and classification based experiments, where the latents are considerably high-dimensional. Here, the explicit models do “learn the correct latents” *as best as possible*, and both Figure 3 and 5 supports this result. The crucial component to understanding this is that the explicit model (even without the oracle prediction function; blue) achieves the same task performance as when the ground-truth latent variable is used to provide an auxiliary supervision signal (purple), which should be read as a performance ceiling. Note that perfect latent variable decoding accuracy for the classification tasks is impossible because with finite contexts there is insufficient evidence to infer the true underlying decision boundary, which accounts for the reviewer’s concerns about Figure 5(a) showing high latent variable decoding error in these plots. Indeed, the models trained to predict true latent variables (purple models) in Figure 5(a) show that the explicit model (blue) decodes the latents just as well across tasks.
>
> [1/3]

---

> > ### Author Response · Authors · 2024-11-21
> > **Author Response**
> >
> > > **Please also provide more evidence that the explicit model learns the task vector better than the implicit one.**
> >
> > We acknowledge the reviewer’s point on comparisons to the implicit model in the ability to learn the task vectors. However, it is an ill-posed problem precisely because it is unclear where the task statistics could be consolidated in a large transformer system, i.e. at which layer and whether it is distributed in the query token or across multiple tokens. This precise problem makes it hard to make conclusions about how well an implicit model does represent the task vector (for eg. Figure 5b highlights this precise problem).
> >
> > However, to address the reviewer’s concerns, we run the counterfactual procedure used in Alchemy on linear regression next. The results (Figure 9) are even stronger than in Alchemy and show that a low-dimensional (10 dimensions) subspace of the explicit bottleneck accounts for almost all of the performance (MSE) of the model, whereas for the implicit model no intervention locations (i.e. layers of the last token of the context) can get even decent counterfactual MSE. This further supports the claim that the explicit model encodes information about the latent variable which is then leveraged to infer the correct prediction.
> >
> > > **There is further little evidence that the explicit model does not also utilize task vectors in some way or another**
> >
> > Could the reviewer clarify their concern? We precisely do show that the explicit model does utilize task vectors, for both downstream prediction and in counterfactual evaluation. What we claim and validate in our setup is that the way it utilizes the task vectors does not generalize in OOD evaluation.
> >
> > > **On the contrary, the results from “In-Context Learning Creates Task Vectors” are still very valid evidence that “implicit” transformers do in fact use task vectors.**
> >
> > We appreciate the reviewer drawing connections of our work with [2]. However, we believe that while [2] shows that it is possible that ICL in LLMs is governed by task vectors as an emergent phenomena, it is unclear whether their findings are primarily an artifact of the task structure and the use of the $\rightarrow$ symbol, which provides a nice token for consolidation of information.  In particular, it is not clear that if the authors remove that symbol, or consider different kinds of tasks, if their observations would still hold. Our results on counterfactual analysis for linear regression suggest that this may not be the case.
> >
> > Further, [2] shows that pre-trained LLMs can potentially use task vectors. It is unclear (without ablating over layers and tokens; especially with overlaps across different layers and tokens) where this task vector is present (notice the variance in their Figure 3) and whether the degradation in performance that they see (Figure 4 in [2]) is because they couldn’t find all the places where the task vector was present or if the task vectors learned in the model are suboptimal. Our work answers that question exactly, i.e., when the models are trained specifically to uncover task vectors, they are still suboptimal.
> >
> > > **The number of tasks you consider is quite small**
> >
> > Figure 2 showcases that we study 10 different tasks with different inductive biases and OOD evaluation protocols to test the hypothesis of whether an explicit model outperforms an implicit one. We do not think that given the number and diversity of our tasks, it should be considered small; but to alleviate the reviewer’s concerns we have run an additional task for compositional generalization and refer the reviewer to the shared global response for details.
> >
> > > **The dimensionality of the data-points, which you set to 1 or 2 (see Appendix) is also very restricted**
> >
> > We refer the reviewer to Figure 6 which does consider much higher dimensional linear regression problems, as well as Figure 2 where the gene knockout and RAVEN progressive matrices dataset are 5000 and 160 dimensional respectively. Even for some low-dimensional experiments, eg. MLP regression and classification, the latent space is high-dimensional even when the observed space is not.
> >
> > [2/3]

---

> > > ### Author Response · Authors · 2024-11-21
> > > **Author Response**
> > >
> > > > **You only consider a single and quite limited way of generating OOD samples for each task.**
> > >
> > > The reviewer might have misinterpreted some of our results since we do consider two very different ways of generating OOD samples. We look at out-of-domain OOD samples for tasks like (non-)linear regression and classification where the query points come from a wider distribution than the ones used during training. We believe that this is the single way that the reviewer is referring to?
> > >
> > > This is not the only version of OOD we test, however: we do consider a form of compositional OOD generalization, where the latents are compositional and only certain subsets / permutations are seen during training while evaluation is done on a different or larger subset. This refers to the experiments on gene knockouts, RAVEN’s progressive matrices as well as our new experiment on mixture of reusable compositional modules.
> > >
> > > Are there any other notions of OOD generalization that the reviewer would like us to conduct experiments on?
> > >
> > > > **Code and Implementation Details**
> > >
> > > We thank the reviewer for bringing this point forward. We have now included the code in our supplementary submission, and will open-source it at a later time. We also refer the reviewer to Appendix B.1 which contains details regarding the model setup, the number of layers and the size of the hidden layers. Additionally, the error bars used in all our plots represent the minimum and maximum quantity for each plot.
> > >
> > > > **What is the evidence to believe in the first place that learning task vectors might improve generalization? And, especially, what is the evidence that not learning it leads to poor generalization?**
> > >
> > > We stress that learning the task vectors is imperative to the success of machine learning models because perfect generalization requires learning of the true underlying function, which is described via the task vectors by definition. That being said, it is a different question on whether learning such a task vector *explicitly* is better than doing so *implicitly*, which is a much more nuanced question that we study in this work.
> > >
> > > We thank the reviewer for their time and constructive feedback and hope that our responses were sufficient in clarifying the excellent questions asked by the reviewer. We are more than happy to address any further questions that arise during this discussion phase.
> > >
> > > **References**
> > >
> > > [1] Von Oswald, Johannes, et al. "Transformers learn in-context by gradient descent." International Conference on Machine Learning. PMLR, 2023.
> > >
> > > [2] Hendel, Roee, Mor Geva, and Amir Globerson. "In-context learning creates task vectors." arXiv preprint arXiv:2310.15916 (2023).
> > >
> > > [3] Kawaguchi, Kenji, et al. "How does information bottleneck help deep learning?." International Conference on Machine Learning. PMLR, 2023.
> > >
> > > [4] Tishby, Naftali, Fernando C. Pereira, and William Bialek. "The information bottleneck method." arXiv preprint physics/0004057 (2000).
> > >
> > > [5] Alemi, Alexander A., et al. "Deep variational information bottleneck." arXiv preprint arXiv:1612.00410 (2016).
> > >
> > > [6] Tishby, Naftali, Fernando C. Pereira, and William Bialek. "The information bottleneck method." arXiv preprint physics/0004057 (2000).
> > >
> > > [7] Hendel, Roee, Mor Geva, and Amir Globerson. "In-context learning creates task vectors." arXiv preprint arXiv:2310.15916 (2023).
> > >
> > > [8] Shalev-Shwartz, Shai, and Shai Ben-David. Understanding machine learning: From theory to algorithms. Cambridge university press, 2014.
> > >
> > > [3/3]

---

> ### Comment · Reviewer_q3Zg · 2024-11-23
> **Response to Rebuttal**
>
> Dear Authors,
>
> some of my points have been clarified through your explanations and references to the literature as well as additional experiments. I therefore raised the score from 3 to 5.
> However, I would still appreciate a more thorough mathematical treatment going alongside your experiments.
>
> Furthermore, releasing the code in the sublementary material is clearly better than not releasing it at all, but still not ideal for reproducibility.

---

> ### Author Response · Authors · 2024-11-25
> **Author Response**
>
> We thank the reviewer for raising their score. However, we are a bit confused by the feedback provided and would like to request for additional information.
>
> **Mathematical Treatment**: We thank the reviewer for their comment but are unclear as to what kind of additional mathematical formalism they are looking for. Could the reviewer clarify what exact formalism beyond the new Appendix E are they implying?
>
> As an additional note, we are of the strong opinion that empirical evidence holds its own place in scientific contributions. In particular, a number of published works also formalize their theory in standard maximum likelihood estimation or empirical analysis of an emergent phenomena, just like us [1-5]. Could the reviewer provide further clarifications?
>
> **Reproducibility**: We feel that we may have misunderstood the reviewer’s feedback. We plan to release our github repository publicly. Beyond experimental details in the Appendix and release of the code in supplementary, what else does the reviewer want us to provide?
>
> We hope that our response has addressed the reviewer's concerns; and we would be happy to provide any further clarifications.
>
> **References**
>
> [1] Hendel, Roee, Mor Geva, and Amir Globerson. "In-context learning creates task vectors." arXiv preprint arXiv:2310.15916 (2023).
>
> [2] Müller, Samuel, et al. "Transformers can do bayesian inference." arXiv preprint arXiv:2112.10510 (2021).
>
> [3] Garg, Shivam, et al. "What can transformers learn in-context? a case study of simple function classes." Advances in Neural Information Processing Systems 35 (2022): 30583-30598.
>
> [4] Zhang, Chiyuan, et al. "Understanding deep learning (still) requires rethinking generalization." Communications of the ACM 64.3 (2021): 107-115.
>
> [5] Mittal, Sarthak, Yoshua Bengio, and Guillaume Lajoie. "Is a modular architecture enough?." Advances in Neural Information Processing Systems 35 (2022): 28747-28760.

---

> > ### Comment · Reviewer_q3Zg · 2024-11-25
> >
> > Dear authors,
> >
> > thanks for providing further clarifications. I like the new graphics in plate notation in the Appendix- however, this is not exactly what I meant.
> > When saying that
> >
> > > First, there are no theoretical results referenced or provided that support the premise that the constrained architecture should generalize better in the first place. The paper would benefit a lot from including mathematical reasoning in that direction. Given that the premise that learning an explicit representation of the task vector is necessary (not sufficient, that’s clear) for generalization is not well-supported in the first place, the results from Figure 2, which refute this premise, cannot be seen as an interesting contribution.
> >
> > I was clearly not referring for mathematical results for their own sake. I certainly agree that this is not necessary for a good paper.
> > I think interpreting my comment in this way misses my point.
> >
> > Let me rephrase:
> >
> > **Given** that there is evidence that learning the right task vectors is important for generalization in in-context learning, your results are a meaningful contribution.
> >
> > My main concern is not even so much that your results do not show the conclusion, but that I do not believe the assumption that that learning task vectors is important for generalization in in-context learning. And not believing the assumption makes your results less meaningful.
> >
> > I am also fully aware that **if** the right latent $\mathbf{z}$ is learned, this easily leads to the Bayes predictor. But the other direction is, i.e. "an optimal predictor needs to learn $\mathbf{z}$ explicitly" is not obvious to me at all. I know that this is pretty much what you try to investigate, but, to formulate it directly: why should this investigation be interesting? (As an analogy: It is not interesting to publish a paper that claims "logistic regression is not the best method for large-scale image classification" --- you can provide vast evidence easily that disproves your hypothesis, however this is nothing new).
> >
> > Could you please explain how your references [3-6] (I think you added Naftali et al. accidentally twice) exactly support the assumption? I also think adding such an explanation would greatly improve the paper.
> >
> > Another option to make your result a valuable contribution at ICLR is if they refute "learning the right task vectors is necessary for generalization" as a **commonly held belief** among ML researchers. However, I personally am sceptical about this as well.
> >
> > However, if the other reviewers would disagree on that, I might consider adjusting my score.
> >
> > Regarding the reproducibility issues, I want to emphasize that releasing the code publicly on Github should be standard practice - and everything else should automatically raise strong suspicions.
> >
> > ### Your References
> >
> > [3] Kawaguchi, Kenji, et al. "How does information bottleneck help deep learning?." International Conference on Machine Learning. PMLR, 2023.
> >
> > [4] Tishby, Naftali, Fernando C. Pereira, and William Bialek. "The information bottleneck method." arXiv preprint physics/0004057 (2000).
> >
> > [5] Alemi, Alexander A., et al. "Deep variational information bottleneck." arXiv preprint arXiv:1612.00410 (2016).
> >
> > [6] Tishby, Naftali, Fernando C. Pereira, and William Bialek. "The information bottleneck method." arXiv preprint physics/0004057 (2000).

---

> > > ### Author Response · Authors · 2024-11-25
> > > **Author Response**
> > >
> > > We thank the reviewer for their response and for providing the additional context around their concerns. It really helped us frame and understand their question better, and we hope that our response below addresses them adequately.
> > >
> > > > **but that I do not believe the assumption that learning task vectors is important for generalization in in-context learning.**
> > >
> > > The reviewer raises a great point! We first want to stress that learning the task vectors is paramount to generalization in in-context learning.
> > >
> > > Let's assume that the optimal prediction function is $g: \mathcal{X} \times \mathcal{Z} \to \mathcal{Y}$, where $\mathcal{X}$ is the input space, $\mathcal{Z}$ refers to the space of task vectors and $\mathcal{Y}$ the output space. Given some observations $\mathcal{D}$ that come from a latent $z$, the optimal model would essentially be as close as possible to $g(\cdot, z)$ in terms of functional forms. Note that the optimal model should be close to $g(\cdot, z)$ and not just any $g(\cdot, z’)$ for some arbitrary z’. Thus, it will have to encode the information about $z$ somehow in its functional form, and our new results on shortcut injection (Figure 8) show that incentivizing shortcuts in solutions that bypass inferring the right latents does degrade performance in OOD evaluation. Whether it is better to encode the information about task latents *implicitly* or *explicitly* is a separate question, which we answer below.
> > >
> > > > **i.e. "an optimal predictor needs to learn z explicitly" is not obvious to me at all.**
> > >
> > > Having clarified that learning the task vectors is important for generalization in in-context learning, we now pivot to the question of needing to learn it *explicitly*. This is precisely the question that we aim to address, and is one of the core contributions of the work. In particular, we exactly show that learning the task vectors *explicitly* is not paramount *on its own* to better generalization (implicit models perform similarly, or explicit models need further access to a *prediction function* with the right inductive biases).
> > >
> > > > **why should this investigation be interesting?**
> > >
> > > We believe that the reviewer answers this question themselves. As they say, the other direction of whether an optimal predictor needs to learn $z$ *explicitly* is not obvious. It is an unanswered question given that there is limited evidence that studies ICL under this regime. Additionally, the reviewer agrees that one side holds, which is that the right latent $z$ *and the prediction function* does lead to Bayes optimal predictor. Then, wouldn’t the reviewer agree that it is an *interesting* and *important* scientific contribution to provide evidence as to whether the converse holds (does an optimal predictor need to learn $z$ explicitly?)? Note that this is different from the logistic regression analogy as there is ample evidence showing that better approaches exist for large-scale image classification; in ICL, such evidence of whether explicitly modeling the task latents is beneficial or not is currently lacking.
> > >
> > > > **Could you please explain how your references [3-6] (I think you added Naftali et al. accidentally twice) exactly support the assumption?**
> > >
> > > The references on information bottleneck [3-6] show that leveraging an information bottleneck by regulating the amount of information in some latent representation can lead to better generalization; which motivates our study of explicit models as they provide an explicit low-dimensional representation to regularize information flow from context to prediction. This is evidenced by the plate diagram (Figure 11) where the explicit model’s prediction $y_*$ is conditionally independent of the context, when conditioned on the bottleneck $z$.
> > >
> > > > **I want to emphasize that releasing the code publicly on Github should be standard practice**
> > >
> > > We completely agree with the reviewer; and that is indeed our goal. We hope that our providing the code in the supplementary ascertains the reviewer that it is what we aim to do.
> > >
> > > We hope that our response addresses the reviewers’ concerns, and we would be happy to provide additional clarifications if needed.

---

> ### Comment · Reviewer_q3Zg · 2024-11-27
>
> Dear authors,
>
> Thank you for your reply.
>
> However, I am not sure if I understand your first point
>
> > We first want to stress that learning the task vectors is paramount to generalization in in-context learning.
>
> Given an optimal $g\_{z} : \\mathcal{X} \\rightarrow \\mathcal{Y}$ that is generated via some $z$, which is the case you consider. (i.e. $g\_{z}$ is your $g$ for fixed $z$). And given a model $f: \\mathcal{X} \\rightarrow \\mathcal{Y}$ that learns to approximate $g\_{z}$.
> I.e. the risk
> $$ R = \\mathbb{E}\_{x \sim P^\\mathcal{D}}[l(g_{z}(x), f(x))] $$ is small for some loss function $l$.
>
> Where does the dependence of $f$ on $z$ come from??? Could you elaborate?
>
> The model $f$ might in fact even learn an arbitrary task vector at some point. To see this, consider a bijection $h: \mathcal{X} \rightarrow \mathcal{Z}$ and define $g'(x) = g_{z}(h^{-1}(h(x)))$. Then $g'$ is still optimal, in fact even $g = g'$, but there is an arbitrary $h(x)$ to be found somewhere in the computation $g'$ does. I.e. what happens if $h$ is the encoder before the information bottleneck and $g_z \circ h^{-1}$ is the decoder after the information bottleneck?!
>
>
> > We believe that the reviewer answers this question themselves.
>
> I do not agree with this statement. I would even like to point out that reviewer nwsg fully aggrees with my point that you do not show something very surprising in your paper:
>
> > However, it remains unclear whether the core question of the study is worth exploring. In my opinion, it is not surprising that implicit models perform as well as (or slightly better than) explicit models, given that much of modern AI progress has already been achieved implicitly. From this perspective, I think the paper(for the future work) could provide more valuable insights by either: (1) Identifying tasks where explicit models outperform implicit models, or (2) Exploring why implicit models perform better than explicit models.
>
> I also do not agree with you saying
>
> >Then, wouldn’t the reviewer agree that it is an interesting and important scientific contribution to provide evidence as to whether the converse holds (does an optimal predictor need to learn explicitly?)?
>
> As I said before, that the "converse" does not hold is not surprising.

---

> > ### Author Response · Authors · 2024-11-28
> > **Author Response**
> >
> > We show below that to minimize risk in our settings, predictors must encode the task latents somewhere. There is no need for the task latent to be identified by the bottleneck, or the implicit model -- bijective transformations of $z$ leave the information intact.
> >
> > > **Where does the dependence from f on z come from??? Could you elaborate?**
> >
> > As the reviewer themselves establishes, let’s say that the $y = g_z(x)$ is the true data generating process. We know that the empirical risk, e.g., with mean squared error as the loss function, is minimized by the true $E[Y|X]$ function, which depends on $z$. Thus, if $f(\cdot)$ minimizes risk, it must contain $z$-relevant information somewhere. One can see that changing either $g$ or $z$ necessitates a change in $f$ for optimality, which shows that $f$ must contain information about *both*.
> >
> > Please note that when we say *encode information*, we do not mean that the model has to use the true representation $z$ -- if it applies a bijection to the true $z$, we don't lose any information about $z$.
> >
> > The setting we study is from the lens of ICL where the model has to infer $z$ from context $\mathcal{D}$ and a similar observation holds. We provide further clarification on why the task vector we learn cannot be arbitrary below.
> >
> > > **Then g′ is still optimal, but there is an arbitrary h(x)**
> >
> > The setup described above is not possible under our explicit model because the decoder operates solely on the query and the information bottleneck (refer to the plate diagram). Thus, if the information bottleneck is something arbitrary, the decoder cannot make correct predictions for the query since the context describes what the predictions should look like. Thus, the information bottleneck would need to encode $z$’s information somehow.
> >
> > That being said, if the information bottleneck encodes a bijection of $z$, that is okay under our setup as it contains all the information about the task latent, even if in some other representation form or space.
> >
> > > **...reviewer nwsg fully aggrees with my point that you do not show something very surprising in your paper**
> >
> > We refer to our corresponding response to reviewer nwsg which highlights that explicit modeling is fundamentally used in a multitude of current settings. Consider standard supervised learning: the parameters that are optimized by learners like SGD or Adam are exactly what we refer to as "task latents" in the ICL context. Thus, supervised learning is about explicitly modeling task vectors and using it to make a prediction for a query independently of the context (i.e., the training dataset). For example, one can see Figure 12 as the case for image classification, where $z$ could represent the parameters of a CNN, $\psi$ the hyperparameters of gradient descent, and $\gamma$ as defining the forward pass of the CNN given the parameters $z$. Given that such explicit modeling is predominant in machine learning, it is not clear whether explicit or implicit modeling is preferable.
> >
> > Even further, our experiments with known prediction functions highlight that with the right inductive biases, the explicit model might be preferable.
> >
> > We additionally highlight related work showing that transformers suffer from shortcuts [1-4], as well as there is evidence of ICL relying on induction heads and non-parametric inference mechanism for predictions [5-8]. Hence it is reasonable to think that biasing against induction-head-like shortcuts could improve generalization by giving rise to other more parametric mechanisms [9-10]. This is in addition to works that show incorporating information bottleneck [11] or inductive biases (e.g. parametric modeling, modularity, attention) can be useful for generalization [12-13].
> >
> > **References**
> >
> > [1] Bihani, Geetanjali, and Julia Taylor Rayz. "Learning shortcuts: On the misleading promise of nlu in language models." arXiv preprint arXiv:2401.09615 (2024).
> >
> > [2] Liu, Bingbin, et al. "Transformers learn shortcuts to automata." arXiv preprint arXiv:2210.10749 (2022).
> >
> > [3] Yang, Sohee, et al. "Do Large Language Models Latently Perform Multi-Hop Reasoning?." arXiv preprint arXiv:2402.16837 (2024).
> >
> > [4] Tang, Ruixiang, et al. "Large language models can be lazy learners: Analyze shortcuts in in-context learning." arXiv preprint arXiv:2305.17256 (2023).
> >
> > [5] Hahn, Michael, and Navin Goyal. "A theory of emergent in-context learning as implicit structure induction." arXiv preprint arXiv:2303.07971 (2023).
> >
> > [6] Wang, Lean, et al. "Label words are anchors: An information flow perspective for understanding in-context learning." arXiv preprint arXiv:2305.14160 (2023).
> >
> > [7] Crosbie, Joy, and Ekaterina Shutova. "Induction heads as an essential mechanism for pattern matching in in-context learning." arXiv preprint arXiv:2407.07011 (2024).
> >
> > [8] Han, Chi, et al. "Explaining emergent in-context learning as kernel regression." arXiv preprint arXiv:2305.12766 (2023).
> >
> > [1/2]

---

> > > ### Author Response · Authors · 2024-11-28
> > > **Author Response**
> > >
> > > [9] Todd, Eric, et al. "Function vectors in large language models." arXiv preprint arXiv:2310.15213 (2023).
> > >
> > > [10] Hendel, Roee, Mor Geva, and Amir Globerson. "In-context learning creates task vectors." arXiv preprint arXiv:2310.15916 (2023).
> > >
> > > [11] Tishby, Naftali, Fernando C. Pereira, and William Bialek. "The information bottleneck method." arXiv preprint physics/0004057 (2000).
> > >
> > > [12] Goyal, Anirudh, and Yoshua Bengio. "Inductive biases for deep learning of higher-level cognition." Proceedings of the Royal Society A 478.2266 (2022): 20210068.
> > >
> > > [13] Andreas, Jacob, et al. "Neural module networks." Proceedings of the IEEE conference on computer vision and pattern recognition. 2016.
> > >
> > > [2/2]

---

> > > > ### Author Response · Authors · 2024-12-02
> > > > **Author Response**
> > > >
> > > > Dear reviewer,
> > > >
> > > > We thank you for your engagement and feedback. As we are nearing the end of the discussion period, we hope that all the concerns have been addressed and would like to request the reviewer for an opportunity to answer any additional questions or doubts that may remain.

---

### Official Review · Reviewer_uCxn · 2024-11-02

**Soundness:** 3
**Presentation:** 4
**Contribution:** 2
**Rating:** 8
**Confidence:** 4

**Summary:**

The authors of this paper propose a hypothesis -- namely, that the in-context learning capabilities of transformers are limited by their tendency to be deceived by shortcuts (as demonstrated in [1]) -- and test it by benchmarking a transformer they introduce to prevent shortcut learning (explicit model) against a standard autoregressive transformer (implicit). The explicit model is a slightly modified architecture in which the parameters that are optimized to perform next-token prediction are not able to attend to specific hidden states, whereas the implicit model can do so. The benchmark is performed over a variety of tasks in in-domain (ID) and out-of-domain (OOD) settings. The results show that the explicit model is not superior to the implicit one on ID data. They show that the bottleneck of the explicit model encodes useful representations of the learned task but the next-token prediction parameters are suboptimal at exploiting it. Additional experiments to investigate the interpretability and scaling properties of the explicit and implicit models are performed.

[1] Large Language Models Can be Lazy Learners: Analyze Shortcuts in In-Context Learning, Tang et al, 2023

**Strengths:**

* **Originality** This paper brings a new perspective to recent work on the problem of shortcut learning. Designing a network to prevent shortcut learning and applying it in the way the authors have done is novel. Framing the investigation in a parametric vs. non-parametric setting is insightful and highlights the operation of attention mechanisms as a particular feature of interest.
* **Quality** This is a high-quality paper. The evaluation on at least five types of tasks is thorough. The authors' use of tasks with known latents indicates a high level of rigor. The authors state the general hypothesis under investigation and provide additional hypotheses (line 165) about which model -- implicit or explicit -- are best suited to the evaluation tasks. The additional experiments (starting line 294) are informative.
* **Clarity** This paper is clear and well written. The setup of the problem and the motivation for the work is communicated concisely.
* **Significance** The contribution of this paper is a novel perspective on a timely problem. Robustness of large transformers is an area of active investigation -- including, shortcut learning (see [1]) -- and this paper shows to an extent that when a network is constrained in a targeted way, generalization does not necessarily improve.

**Weaknesses:**

While the overall quality of the paper is quite good, much of the quality can be said to be tactical (i.e. the execution) rather than strategic (i.e. the motivation and contribution). In tactical terms, the paper is beautiful. Strategically, it can be improved.

While I acknowledge and appreciate the principled argument in favor of the explicit model in the context of parametric vs non-parametric learning (see p. on line 70, Section 3, and Figure 1), I believe the paper would be very much improved if it verified empirically that the explicit model is itself resistant to shortcut learning in the presence of **injected** shortcuts. Whereas [1] injects shortcuts ("triggers") into data in an ICL setting, the work presented here appears to use datasets without such injections. Creating, if necessary, a synthetic dataset for evaluating robustness to shortcuts, and empirically demonstrating that the explicit model is resistant to shortcuts would strengthen the paper by showing a method by which existing large-scale models might be made resistant to shortcuts. Further showing that the explicit model exhibited consistent resistance to shortcuts as the model size increases would be a substantial contribution. Finally, showing this would provide solid empirical ground on which the argument in p. on line 70 rests.

I would prefer that the authors removed "commonly" from their conclusion (line 512). It's non-controversial that shortcuts are a likely explanation for well-demonstrated non-robustness of transformers to incidental variations of their inputs. In the context of the sentence on line 512, however, the use of "commonly" can be taken to imply that most people believe that shortcut learning is the reason for poor OOD generalization. If that particular view w.r.t. OOD generalization is commonly held, I would expect there to be be some literature -- particularly since [1] was published in 2023 -- claiming exactly that. If it exists, it should be cited in the introduction.

Nit: The classification tasks in the first row of Figure 4 are missing the label on the Y axis.

**Questions:**

1. Do you have results comparing the explicit model's performance to the implicit model on a task in which triggers have been deliberately injected? If you do, please report them here. If they show that the explicit model is robust to shortcuts, I would find the result compelling.
2. Beyond what you argue in the paragraph starting on line 70, does your understanding of OOD generalization include robustness to variation of incidental features on ID data?

---

> ### Author Response · Authors · 2024-11-21
> **Author Response**
>
> We thank the reviewer for their valuable and detailed comments and are glad that they thought our work “brings a new perspective”, is “high quality” and brings a “novel perspective in a timely manner”. We now address their key questions and concerns.
>
> >  **I believe the paper would be very much improved if it verified empirically that the explicit model is itself resistant to shortcut learning in the presence of injected shortcuts.**
>
> The reviewer brings up an interesting and useful suggestion of studying how sensitive the explicit model is in the presence of injected shortcuts. To test this, we conduct an additional experiment based on sinusoidal regression where the queries were by design sampled close to the context during training. Our results highlight that the implicit model led to worse OOD generalization far from the context than the explicit model, highlighting that it was leveraging kernel-regression style shortcuts as opposed to learning the underlying sinusoid function. We refer the reviewer to the shared global response for details about the experiment and the results.
>
> > **the use of "commonly" can be taken to imply that most people believe that shortcut learning is the reason for poor OOD generalization.**
>
> We refer the reviewer to works [1-5] which highlight how transformer models (and pre-trained large language models) fail to generalize because they rely on shortcuts as opposed to learning the underlying task structure. Since we study ICL in the context of transformer models, the kind of shortcuts preventing generalization studied in [1-5] transfer to our setting as well. More mechanistically, recent works [6-8] have identified pattern matching mechanisms like induction head to underlie ICL; something that further motivates our study of explicit models.
>
> Based on the related work, we felt that this was a commonly held belief, but if the reviewer feels that it is not the case, we would be happy to omit it.
>
> > **does your understanding of OOD generalization include robustness to variation of incidental features on ID data?**
>
> It is unclear to us what the reviewer means by “incidental features”. Could they please clarify?
>
> We thank the reviewer for their time and constructive feedback and hope that our responses were sufficient in clarifying the excellent questions asked by the reviewer. We are more than happy to address any further questions that arise during this discussion phase.
>
> **References**
>
> [1] Anil, Cem, et al. "Exploring length generalization in large language models." Advances in Neural Information Processing Systems 35 (2022): 38546-38556.
>
> [2] Zhang, Yi, et al. "Unveiling transformers with lego: a synthetic reasoning task." arXiv preprint arXiv:2206.04301 (2022).
>
> [3] Liu, Bingbin, et al. "Transformers learn shortcuts to automata." arXiv preprint arXiv:2210.10749 (2022).
>
> [4] Yang, Sohee, et al. "Do Large Language Models Latently Perform Multi-Hop Reasoning?." arXiv preprint arXiv:2402.16837 (2024).
>
> [5] Bihani, Geetanjali, and Julia Taylor Rayz. "Learning shortcuts: On the misleading promise of nlu in language models." arXiv preprint arXiv:2401.09615 (2024).
>
> [6] Crosbie, J., & Shutova, E. (2024). Induction heads as an essential mechanism for pattern matching in in-context learning. arXiv preprint arXiv:2407.07011.
>
> [7] Tang, R., Kong, D., Huang, L., & Xue, H. (2023). Large language models can be lazy learners: Analyze shortcuts in in-context learning. arXiv preprint arXiv:2305.17256.
>
> [8] Wang, L., Li, L., Dai, D., Chen, D., Zhou, H., Meng, F., ... & Sun, X. (2023). Label words are anchors: An information flow perspective for understanding in-context learning. arXiv preprint arXiv:2305.14160.

---

> ### Comment · Reviewer_uCxn · 2024-11-21
>
> Based on your recent results, and your overall response, I have updated my score from 6 to 8. I would very much appreciate it if the authors abbreviated parts of the paper and included the results on injected shortcuts (Figure 8) and the update to Appendix D in response to "discussion on how observations differ across tasks" (response to reviewer nwsg).
>
> ## Injected shortcuts
>
> Thank you for acting on my suggestion. This result provides some empirical support for your presentation of the explicit model as parametric and the implicit model as non-parametric (p. on line 70, Section 3, and Figure 1). (To be fair to readers, please use the same range for the Y axis in both subfigures of Figure 8.) An additional improvement would be to demonstrate this on a few more tasks.
>
> ## OOD generalization and "commonly".
>
> Thank you for the additional references. Please cite these to support your framing.
>
> Part of what I'm referring to wrt "commonly" is that generically, under empirical risk minimization, distributional assumptions hold, and there's no guarantee that a model will generalize beyond the distribution on which it was trained. I believe it's fair to argue that the brittleness of large transformers is, in part, due merely to their vast over-parameterization. They're high-variance models, although ones that happened to have been trained on a large amount of data. Please witness, e.g., how in Tang et al, 2023, the tendency of transformers to be susceptible to injected shortcuts increased with larger model. This suggests that the the particular shortcut behavior demonstrated in Tang et al, 2023, can be attributed in part to the sheer over-parameterization of the model(s).
>
> That said, I believe it would help your work be properly understood in relation to other work if you were to state explicitly which kinds of shortcuts your work does not address. As you state in your response to reviewer q3Zg, you're addressing "kernel-regression / nearest neighbor style shortcuts". This is important, because it's possible that shortcuts demonstrated in some of the references you provided here in your response are of a different style altogether. You could mention in your framing that (1) you're addressing parametric vs. non-parametric styles of shortcuts and (2) not addressing shortcuts due to e.g. over-parameterization. That's just one example. One question is whether the [1-6, 8] you shared contain other styles of shortcuts. Even a rough and heuristic categorization of the styles of shortcuts could help readers to know in what context to understand your work.
>
> ## Robustness to variation of incidental features on ID data
>
> Apologies for my infelicitous phrasing. I asked this question to stimulate discussion to better understand your reasoning about OOD generalization (cf. my comment about ERM above). Robustness is the opposite of brittleness. In Tang et al, 2023, to continue with that reference, the injected shortcuts (such as the use of "movie" in their Figure 2) are variations that are incidental to the task. So I suspect that your answer to my question is affirmative, in a sense. Where to place the line between robustness to (noisy) ID data and OOD data can be argued in a different venue.

---

> > ### Author Response · Authors · 2024-12-02
> > **Author Response**
> >
> > We appreciate the reviewer’s feedback and increase in score; and are especially happy that our response addressed their concerns. We also apologize for the late response.
> >
> > We will make changes to our draft to include additional discussion in Appendix D regarding shortcut injection as well as cite the above-mentioned literature as additional support regarding shortcuts. Following the reviewer’s advice, we will also make it clearer in our introduction that the kind of shortcuts we focus on are nonparametric, kernel-regression based shortcuts which are different compared to overparameterization based ones, as the reviewer rightfully points (e.g. Tang et. al). The primary contribution of our work is to study such shortcuts in controlled and systematic manner; unlike Tang et. al which conducts analysis on LLMs which, even though is extremely relevant, could be riddled with more confounders underlying their findings.
> >
> > We once again thank the reviewer for their time and feedback.

---

### Official Review · Reviewer_nwsg · 2024-11-04

**Soundness:** 3
**Presentation:** 2
**Contribution:** 2
**Rating:** 6
**Confidence:** 3

**Summary:**

The main goal of this paper is to study the relationship between learning latent variables and in-context learning performance. To do so, the paper compares the in-context learning abilities of implicit and explicit models, where the explicit model has an architecture with a bottleneck designed to explicitly infer the task latents. Furthermore, the paper examines how performance changes in explicit models causally through various ablation studies.

**Strengths:**

1. The paper clearly describes the procedures and results of the experiments.
2. The paper incorporates various tasks, ranging from simple regression tasks to more realistic tasks such as gene targeting tasks.
3. The paper studies causal relationships between ICL performance and various factors, such as the presence of an auxiliary ground-truth task latent loss.

**Weaknesses:**

The paper addresses an interesting question, but the robustness of its implications remains unclear. The finding that explicitly learning latent variables does not enhance performance may be limited to the specific tasks chosen for evaluation, especially those where implicit models already perform well.
Also, the paper lacks a discussion on how observations differ across tasks. For example, in Figures 3 and 4, the performance trends (accuracy gains) of the explicit model and other factors vary inconsistently across task types, yet the paper does not explore how or why these results differ by task.

**Questions:**

Is it possible to study the task-specific relationships shown in Figures 3 and 4 in more depth? For example, could you provide a detailed analysis of why certain tasks benefit from the explicit model while others do not?
In Figure 4b, can you explain why MLP predictions are generally less effective than those from transformers?

---

> ### Author Response · Authors · 2024-11-21
> **Author Response**
>
> We thank the reviewer for their detailed comments and are glad that the reviewer found our work “addressing an interesting problem” and clearly written with a diversity of tasks. We take this opportunity to further clarify the concerns that were raised by the reviewer.
>
> > **The finding that explicitly learning latent variables does not enhance performance may be limited to the specific tasks chosen for evaluation, especially those where implicit models already perform well**
>
> We thank the reviewer for bringing this point up and clarify that we actually tried to choose tasks for which we intuitively felt that the explicit model would do better, and not vice versa. For example, for all our tasks with low-dimensional task latents, eg. sinusoidal regression, linear regression, linear classification, raven’s progressive matrices, gene targeting and alchemy, the latents are actually low-dimensional and thus intuitively the explicit model should do better. However, counter-intuitively we see that the implicit model performs just as well, and often better; which really highlights the strength of our results supporting the hypothesis. However, to alleviate the reviewer’s concerns, we conduct an additional experiment based on a modular and compositional task, the results and details of which are provided in the shared global response.
>
> > **discussion on how observations differ across tasks**
>
> We thank the reviewer for bringing this point forward and have now revised the draft and presented the requested analysis in Appendix D. Below we explain differences in results across tasks:
>
> *Classification tasks vs. regression tasks.* OoD performance is generally strong (across all models) for classification because decision boundaries are within the training domain and do not change beyond it. In contrast, for regression tasks, the function continues to change beyond the observed training domain, making OoD prediction more difficult. This is also why known prediction functions give little benefit in classification tasks: they are already solved well OoD with ordinary implicit and explicit models.
>
> *The explicit model with known prediction function does not give benefits in nonlinear (MLP) regression.* This is because the problem of inferring an MLP’s weights given some context examples is too difficult, so the explicit model potentially opts for a different, non-parametric solution. This is supported by the latent variable decoding results in Fig. 5 (previously Fig. 4), which show that even with a known prediction function the explicit model does not learn to infer the correct latent variable for the nonlinear (MLP) regression task.
>
> > **Figure 4b, can you explain why MLP predictions are generally less effective than those from transformers?**
>
> We do not have a very clear answer as to why MLP predictions are generally less effective than those from transformers, and can only conjecture that it has to do with the inductive bias of the transformer decoder, in particular of the attention modules and dot-product similarity metric. These results are aligned with our point that the functional form of the decoder in the explicit model is crucial to reap the benefits of the learned latents.
>
> We thank the reviewer for their time and constructive feedback and hope that our responses were sufficient in clarifying all the useful questions asked by the reviewer. We are more than happy to address any further questions that arise during this discussion phase.

---

> > ### Author Response · Authors · 2024-11-25
> > **Request for Discussion**
> >
> > Dear reviewer,
> >
> > We are very appreciative of your time and feedback. As we are nearing the end of the rebuttal period, we would like to request the reviewer for an opportunity to answer any additional questions or doubts that may remain.

---

> ### Comment · Reviewer_nwsg · 2024-11-25
>
> Thank you for your response, which partially addressed my previous questions. I believe the additional tasks and discussions significantly enhance the paper. However, it remains unclear whether the core question of the study is worth exploring. In my opinion, it is not surprising that implicit models perform as well as (or slightly better than) explicit models, given that much of modern AI progress has already been achieved implicitly. From this perspective, I think the paper(for the future work) could provide more valuable insights by either: (1) Identifying tasks where explicit models outperform implicit models, or (2) Exploring why implicit models perform better than explicit models.

---

> > ### Author Response · Authors · 2024-11-26
> > **Author Response**
> >
> > We thank the reviewer for their feedback and for engaging in the discussion. We provide additional context around why the problem we study is worth pursuing and hope that our response below alleviates the reviewer’s concerns.
> >
> > >  **it is not surprising that implicit models perform as well as (or slightly better than) explicit models**
> >
> > We acknowledge the reviewer’s insights and provide an alternative viewpoint. A large part of ML community *even currently* leverages explicit models for various modeling tasks. For example, tasks like image classification (e.g. MNIST or CIFAR classification; as well as other modeling tasks) are solved by parameterizing an explicit model $f_\theta$ through a MLP / CNN / ViT. We call this an explicit approach primarily because given $f_\theta$, the prediction of a query is conditionally independent of the observations (refer to the plate notation for explicit model in Figure 12). This is in opposition to more implicit style nonparametric solutions like nearest neighbors or its deep-learning counterparts.
> >
> > Given that actually there is considerable evidence in support of the explicit approach (without bringing in-context learning to the mix), it was not obvious to us that implicit models should outperform the explicit counterparts. Even further, we found such a study to be lacking in literature for ICL, where the lines between non-parametric and parametric models is more blurry as the network can operate on arbitrarily sized contexts but with a parametric processing machinery in the mix.  Further, we want to remind the reviewer that our result is not solely that explicit models underperform implicit ones; we argue that the downstream prediction is a cause for failure of explicit models, thus motivating further research.
> >
> > > **Identifying tasks where explicit models outperform implicit models**
> >
> > Our findings on counterfactual analysis (Figures 5b and 9) and shortcut injection (Figure 8) show precisely this; that while generally implicit models outperform explicit ones there are cases where the latter are preferable. In particular, when the tasks contain extreme shortcuts or when one needs to make counterfactual predictions, the implicit model suffers considerably more because it relies (to varying degrees) on kernel-regression based shortcuts.
> >
> > > **Exploring why implicit models perform better than explicit models.**
> >
> > Implicit models are inherently more flexible than explicit models in terms of expressivity, as the latter is constrained by the bottleneck $z$. Whether this bottleneck aids or hinders generalization is an open topic of research which has been studied in different settings [1-3]. This hypothesis has been under-explored in ICL, and our aim was to conduct systematic experiments to validate it under a diversity of tasks. Our results indicate exactly why the implicit model is better than the explicit one: the latter fails to leverage the task latents correctly for out of distribution prediction.
> >
> > **Additional Experiment**: To further motivate the problem and provide further valuable insights about the problem, we conduct an additional experiment where we augment the explicit model such that the task latents $z$ also depends on the query $x_q$; we refer to this as the implicit proxy model. This essentially makes the model implicit in nature while maintaining the exact same parameterization as the explicit model. Our new results on linear and sinusoidal regression in Figure 11 indicate that in out-of-distribution (as we move away from $0$ on the x-axis), the implicit proxy model outperforms the explicit model (as we already see often between implicit and explicit model); however, the prediction of the task latents in the implicit proxy model gets worse, i.e. follows the opposite trend. We refer the reviewer to Figures 11-12 and Appendix E for further details, and will incorporate these results to provide further analysis in Appendix D.
> >
> > We hope that our response addresses the reviewers’ concerns, and we would be happy to provide additional clarifications if needed.
> >
> > **References**
> >
> > [1] Kawaguchi, Kenji, et al. "How does information bottleneck help deep learning?." International Conference on Machine Learning. PMLR, 2023.
> >
> > [2] Tishby, Naftali, Fernando C. Pereira, and William Bialek. "The information bottleneck method." arXiv preprint physics/0004057 (2000).
> >
> > [3] Alemi, Alexander A., et al. "Deep variational information bottleneck." arXiv preprint arXiv:1612.00410 (2016).

---

> > > ### Comment · Reviewer_nwsg · 2024-12-01
> > >
> > > Thank you for addressing my follow-up concerns. I am convinced that the novelty of the paper lies in its first exploration of nonparametric vs. parametric modeling for ICL. Also, the additional experiment about shortcut is impressive and clear. As a result, I have increased my score to 6.
> > >
> > > However, I still question whether the paper’s definition of an explicit model—characterized as a model constrained by a low-dimensional task latent—is fully appropriate for addressing the proposed question. Specifically, as noted in the authors' responses, there can be varying levels of parameterization, such as MLPs, CNNs, or ViTs. Consequently, the conclusions about parameterization versus nonparameterization can heavily depend on the degree of parameterization as well as the tasks being studied. Since the explicit model proposed in this work represents a specific or simplified form of parameterization, I believe conclusions like “explicit modeling does not benefit ICL” may be overstated if based solely on particular parameterization. I would appreciate it if the authors could address these points in more detail.

---

> > > > ### Author Response · Authors · 2024-12-02
> > > > **Author Response**
> > > >
> > > > We thank the reviewer for their response and for increasing their score. We completely agree with the reviewer that the conclusions can depend heavily on the degree of parameterization and the tasks studied. We answer both of these points below:
> > > >
> > > > **Degree of Parameterization**: We had performed preliminary analysis to study this by ablating over the size of the latent bottleneck. Our experiments indicated minimal performance difference when making the latent bottleneck bigger than what we currently consider. Even further, explicit models like MLPs, CNNs, etc. perform extremely well *when trained on a large corpus of data*, and handling such a large context length through an in-context learner has its own set of computational challenges that are outside the scope of our work.
> > > >
> > > > **Tasks Studied**: We agree that the nature of the tasks play a role which is exactly why we conducted experiments on a varied suite of ten different tasks with different kinds of underlying inductive biases.
> > > >
> > > > While the reviewer brings up important points regarding the parameterization (e.g. MLPs, CNNs, or ViTs), we would like to point them to additional inductive biases that potentially lead to such explicit models performing well, e.g. -- gradient based learner like Adam or SGD, ability to handle large dataset size (gradient based learners can handle extremely large datasets but in-context learners are limited by context length), similar computational advantages, etc. We believe that such analysis would be very relevant future work, but outside our scope since our primarily aim is to analyze explicit models *for ICL*. We will make the corresponding updates to our draft to reflect this more clearly, and thank the reviewer for helping us improve our work.

---

### Official Review · Reviewer_gSWz · 2024-11-05

**Soundness:** 2
**Presentation:** 1
**Contribution:** 1
**Rating:** 3
**Confidence:** 3

**Summary:**

A commonly believed hypothesis is that Transformers do ICL through brittle statistical shortcuts rather than by inferring the underlying generative latent variables of the task, and that this explains their inability to generalize outside of the training distribution.
This paper attempts to understand if it is true. They minimally modify the Transformer architecture with a bottleneck designed to prevent shortcuts in favor of more structured solutions, and then compare performance against standard Transformers across various ICL tasks. Their empirical simulations suggest that little discernible difference exist between the two different approaches, thus questioning the previous belief.

**Strengths:**

The strength of this paper is that they provide empirical simulations on their claim. Their plot looks very carefully drawn. The problem seems to be interesting.

**Weaknesses:**

The major weakness of this paper is its presentation, which is not very clear to the reviewer. In particular, the reviewer felt it very difficult to understand (1) The contributions of this work. (2) The structure and logic of this work. (3) The motivation of this work. Given these concerns, the reviewer believes that the current paper requires significant rewrite to make it publishable at ICLR.

In particular, the author has not highlighted their key contributions in this work at the abstract/introduction part. The motivations seem like a course project rather than a publishable work.

**Questions:**

Given the above weakness, the reviewer believes that the author needs to clarify (1) what is the contribution and novelty in this work. (2) how is the claim being rejected by the experiments. (3) whether the experiments are scalable to LLMs  (4) what is the motivations in this work and the impact of this work to the research communities.

And at the current stage, the reviewer believes a significant improvement and rewrite is needed to get it publishable.

---

> ### Author Response · Authors · 2024-11-21
> **Author Response**
>
> We thank the reviewer for their feedback, but it is not clear from their review as to what parts of the work were unclear. We provide a comprehensive assessment of the positioning, motivation and contribution of our work below, as well as an overview of our empirical analysis. We hope that it clarifies the reviewer’s concerns, and would be happy to address any further comments.
>
> > **Motivation**
>
> Multiple works [6,7] claim that transformers perform ICL through pattern matching mechanisms such as induction heads, acting as shortcuts. Our motivation is to test if minimally modifying the transformer architecture to incentivize it to explicitly learn the task latents would prevent learning of shortcuts and lead to better OOD generalization. We test this theory by analyzing whether such a model actually learns the latents well, if it generalizes better than the implicit model and identify why it doesn’t.
>
> > **Contributions and Novelty**
>
> We have now revised the draft to include a list of our contributions at the end of the introduction. The primary contribution of our work is that learning task latents correctly is not necessarily the only or biggest problem for generalizing in-context to out-of-distribution queries and tasks. This finding provides a different viewpoint to the importance and relevance of inferring the right latents (in contrast to [4,5]) and shows that even if this part is solved, it does not guarantee good OoD performance. Note that this negative finding is important to the field, as it significantly prunes the search space of solutions [3].
>
> > **Impact**
>
> We argue that our findings also suggest promising directions. For example, it can lead to future work on improving the prediction module as opposed to context aggregation if OoD generalization is the goal (for instance, through task-specific architectures). Note that this approach is impossible using an “implicit” model, as context aggregation and prediction are entangled and we cannot use different inductive biases for each. Hence, we believe that even though we don’t develop these solutions in our work, our findings and analysis provide a thorough and comparative study to guide future research in OoD generalization for ICL.
>
> > **Structure and Logic**
>
> While other reviewers found our work to be clearly written, we may have been unclear at some places. Could the reviewer clarify what in particular about the structure and logic was unclear, so that we can both rectify it in our draft and clarify it to the reviewer?
>
> > **motivations seem like a course project rather than a publishable work**
>
> In this work, we study whether learning the right functional form of the solution through task latents leads to better OOD generalization in In-Context Learning, which as the reviewer themselves have pointed out is an “interesting” problem. We believe that our controlled experiments and systematic analysis provides answers and novel insights to an important question, which is valuable to science and beyond the scope of a course project.
>
> > **how is the claim being rejected by the experiments**
>
> Our claims are being rejected by the experiments in the following manner
>
> *Learning shortcuts prevents OOD generalization*: If learning shortcuts over the right parametric inference procedure was the main issue preventing OOD generalization in ICL, then the explicit transformer model should perform better than the implicit one. Our experiments show that this is not the case, even though explicit models do learn the true latent variables.
>
> *Failure of explicit models is due to suboptimal downstream prediction*: Our experiments identify that the learned prediction function is unable to combine even the correct latents in a systematic way (Figure 3). In fact we know that the explicit model often does learn a linear representation of these right latents (Figure 5). This identifies the prediction problem as an important one for further research.
>
> [1/2]

---

> ### Author Response · Authors · 2024-11-21
> **Author Response**
>
> > **whether the experiments are scalable to LLMs**
>
> We also emphasize that the questions we investigated cannot be studied in complex modalities like language or vision because the true latents are generally unknown in these settings. Access to the true latents was necessary for the systematic analysis and results in Figure 3 and Figure 5, for instance, along with the insights they provided. It is exactly for this reason that we, as well as a number of related works that explore such questions [1,2], operate in this regime.
>
> While understanding shortcut-learning and robust generalization would be interesting in pre-trained large language models, we consider it as relevant future work that is outside of our current scope. This is primarily because we learn less about the problem by directly evaluating on a modality where neither the nature of shortcuts nor the true latents are known. Naively testing in this domain can lead to conflicting results and evidence, as well as overconfidence in incorrect claims. Thus, our goal is to study these questions in a more controlled setting where we can make claims based on our results with more assurances.
>
> We thank the reviewer for their time and feedback, and hope that our response clarifies their concerns. We would appreciate it if the reviewer would also let us know their specific concerns in more detail to give us a chance to improve our work and provide potential clarifications.
>
> **References**
>
> [1] Von Oswald, Johannes, et al. "Transformers learn in-context by gradient descent." International Conference on Machine Learning. PMLR, 2023.
>
> [2] Garg, Shivam, et al. "What can transformers learn in-context? a case study of simple function classes." Advances in Neural Information Processing Systems 35 (2022): 30583-30598.
>
> [3] Karl, Florian, et al. "Position: Embracing Negative Results in Machine Learning." arXiv preprint arXiv:2406.03980 (2024).
>
> [4] Xie, Sang Michael, et al. "An explanation of in-context learning as implicit bayesian inference." arXiv preprint arXiv:2111.02080 (2021).
>
> [5] Todd, Eric, et al. "Function vectors in large language models." arXiv preprint arXiv:2310.15213 (2023).
>
> [6] Crosbie, J., & Shutova, E. (2024). Induction heads as an essential mechanism for pattern matching in in-context learning. arXiv preprint arXiv:2407.07011.
>
> [7] Tang, R., Kong, D., Huang, L., & Xue, H. (2023). Large language models can be lazy learners: Analyze shortcuts in in-context learning. arXiv preprint arXiv:2305.17256.
>
> [2/2]

---

> > ### Author Response · Authors · 2024-11-25
> > **Request for Discussion**
> >
> > Dear reviewer,
> >
> > We are very appreciative of your time and feedback. As we are nearing the end of the rebuttal period, we would like to request the reviewer for an opportunity to answer any additional questions or doubts that may remain.

---

### Official Review · Reviewer_P9Tn · 2024-11-10

**Soundness:** 3
**Presentation:** 3
**Contribution:** 3
**Rating:** 6
**Confidence:** 2

**Summary:**

The paper studies an important problem of whether learning good latent representation improve in context learning. For that, the authors add a bottleneck to Transformers to encourage structured predictions. Experimental results are presented. Interestingly, the experiments show that learning good latent representation does not help with out of distribution predictions nor downstream tasks.

**Strengths:**

- It is important for the community to learn about such work.
- The problem studied is interesting.

**Weaknesses:**

See below

**Questions:**

The findings from the paper are interesting, do you think that the controlled experiments are enough to draw such a strong conclusions? Or will there be cases where learning good latent representation improves the prediction performance?

---

> ### Author Response · Authors · 2024-11-21
> **Author Response**
>
> We thank the reviewer for their time and feedback, and are glad that they thought that the problem we study is “interesting” and “important for the community”. We address all of their questions below.
>
> > **do you think that the controlled experiments are enough to draw such a strong conclusion?**
>
> We acknowledge the reviewer’s point and think that our controlled experiments are *exactly* what allows us to draw such a strong conclusion: that learning good latents only leads to better ICL performance when paired with an appropriate prediction function. Specifically, knowledge of the ground truth latents is crucial to most of our experiments; something we would not have in less controlled tasks. Nonetheless, we believe our claim is still important (if anything, more) in more natural forms of ICL, where the latents and prediction functions are simply much more complex.
>
> We also conduct additional experiments to test for compositional generalization in a reusable mixture-of-experts task as well as analysis of the proposed models in the presence of shortcuts. We refer the reviewer to the shared global response for details about this setup and the corresponding results. Are there any specific additional experiments that would help solidify the validity of our conclusions for the reviewer?
>
> > **will there be cases where learning good latent representation improves the prediction performance?**
>
> The reviewer brings up a very relevant question. We clarify that one of the conclusions of our empirical evidence is that learning good latent representations *does* in fact help in-context learning, but only when the architecture is able to use them optimally for the downstream prediction. Our experiments with explicit models indicate that the learned downstream prediction ended up being sub-optimal, and replacing it with the “known” prediction function instead (Figure 3) led to strong OOD performance.
>
> We thank the reviewer for their time and constructive feedback. We hope that our responses were sufficient in clarifying all the great questions asked by the reviewer and are more than happy to address any further questions that arise during this discussion phase.

---

> > ### Author Response · Authors · 2024-11-25
> > **Request for Discussion**
> >
> > Dear reviewer,
> >
> > We are very appreciative of your time and feedback. As we are nearing the end of the rebuttal period, we would like to request the reviewer for an opportunity to answer any additional questions or doubts that may remain.

---

### Author Response · Authors · 2024-11-21
**Author Response**

We thank the reviewers for their detailed and insightful comments about our work and are glad that they found our work to be “*important for the community*” (R P9Tn), “*addressing an interesting problem*” (R nwsg), and “*bringing a novel perspective*” (R uCxn). We are encouraged by this positive outlook towards our work and address the shared concerns of the reviewers in this response. We begin by a general overview of changes before a point-by point rebuttal.

Our core motivation and contribution is a systematic investigation of the hypothesis that uncovering true task latents during ICL would lead to better OOD generalization through controlled experiments. We believe that investigating ICL in sequence models and its generalization properties is an important contribution to the literature. Following the reviewers’ suggestions,  we have now explicitly incorporated a list of our contributions at the end of the introduction and moved the related works section to the Appendix.

**Empirical Evidence**: To also alleviate the reviewers’ concerns regarding empirical evidence, we clarify that we validate our hypothesis on a suite of 10 different tasks ranging from (a) regression to classification, (b) synthetic to real world (e.g. linear regression to gene knockouts), (c) underlying parametric to nonparametric function space (e.g. sinusoidal regression to Gaussian Process), and (d) low-dimensional (linear regression: 1-D) to high-dimensional (gene knockouts: 5000-D) experiments; with evaluation ranging from out of domain to compositional generalization.

Even further, we conduct two additional experiments to test for compositional generalization and generalization in the presence of extreme shortcuts. Our new analysis further solidifies the hypothesis that explicit models do not outperform implicit models generally, while additionally showcasing that in some cases of extreme shortcut injection, an explicit model has the capacity of outperforming implicit models. We describe these two experiments in detail below.

**Compositional Generalization in Reusable Modular Mixture of Experts (MoEs)**: We conduct additional experiments on a task that allows decomposition into smaller reusable components which can be composed in a flexible manner. In particular, this task is characterized by some basis functions which can be reused at any layer of application, and the order of their application is governed by the task latent variable.

In particular, this task considers the true prediction function $y_* = g(x_*, z)$ such that $z$ is a 5-dimensional categorical vector, and $g$ has a structured representation. In particular, $y_* = g_{z_5} \circ g_{z_4} \circ \ldots \circ g_{z_1} \left(x_*\right)$ where each $g_i$ is a linear network with a tanh activation function. This induces a modular reusable structure in the task with a low-dimensional latent. The latent controls the order in which different components are applied to the input, and $g_i$ denotes the components themselves. During training, we show only a subset of combinations to learn from but then perform evaluation on all combinations. We revise our draft to describe this setup in Section 4 and Appendix B. Our experiments in Figure 4 indicate that even with this modular and compositional task structure, the implicit transformer outperforms explicit models across various percentages of combinations seen during training.

**Generalization with Shortcut Injection**: Reviewer uCxn brought up an important point about investigating the possibility of certain shortcut solutions. To address this, we perform an additional experiment where during training, we ensure that the query points $x_*$ are sampled extremely close to the context examples. This incentivizes the implicit model to rely more on nearest-neighbor based lookup as opposed to the explicit model, which still needs to distill the information present in the context into a task vector, a process which is independent of the query. After training, we perform evaluation by sampling query points far from the context. Our results in Figure 8 indicate that in such forms of shortcut injection during training, the implicit model’s performance deteriorates considerably while the explicit model still leads to a reasonably good performance; showing the resistance to certain kinds of shortcuts in explicit models which is designed to learn task-specific latents in a bottleneck without access to the query.

We would like to again thank all the reviewers for their valuable time and effort in reviewing our manuscript and are especially grateful for the improvements to our paper that they led to. We hope that our response has addressed all of their concerns, and we would be happy to provide further clarifications if needed.

---

### Meta-Review · Area_Chair_VLwu · 2024-12-20

**Metareview:**

The paper explores the question of whether learning task vectors explicitly or implicitly is more beneficial for generalization in in-context learning, presenting a series of experiments that compare the performance of explicit and implicit models on various tasks. However, despite the paper's interesting insights and experiments, the reviewers have raised significant concerns about the paper's contribution, arguing that the premise that learning task vectors is important for generalization is not well-supported and that the assumption that an optimal predictor needs to learn task vectors explicitly is not obvious. The authors have responded to these concerns with additional arguments and clarifications. However, after discussion with the authors and among themselves, the reviewers find the paper to still be very borderline, with three reviewers leaning towards acceptance and two towards rejection. We will therefore need to reject the paper in its current form. We would still like to encourage the authors to resubmit an improved version of the paper in the future.

**Additional Comments On Reviewer Discussion:**

see above

---

### Decision · Program_Chairs · 2025-01-22

Reject